# BACKPROPAGATION THROUGH COMBINATORIAL ALGORITHMS: IDENTITY WITH PROJECTION WORKS

**Subham Sekhar Sahoo**[*]
Cornell University
Ithaca, USA
*ssahoo@cs.cornell.edu*

**Anselm Paulus**[*]
MPI Intelligent Systems
Tübingen, Germany
*anselm.paulus@tue.mpg.de*

**Marin Vlastelica**
MPI Intelligent Systems
Tübingen, Germany
*marin.vlastelica@tue.mpg.de*

**Vít Musil**
Masaryk University, FI
Brno, Czech Republic
*musil@fi.muni.cz*

**Volodymyr Kuleshov**
Cornell Tech
NYC, USA
*kuleshov@cornell.edu*

**Georg Martius**
MPI Intelligent Systems
Tübingen, Germany
*georg.martius@tue.mpg.de*

## ABSTRACT

Embedding discrete solvers as differentiable layers has given modern deep learning architectures combinatorial expressivity and discrete reasoning capabilities. The derivative of these solvers is zero or undefined, therefore a meaningful replacement is crucial for effective gradient-based learning. Prior works rely on smoothing the solver with input perturbations, relaxing the solver to continuous problems, or interpolating the loss landscape with techniques that typically require additional solver calls, introduce extra hyper-parameters, or compromise performance. We propose a principled approach to exploit the geometry of the discrete solution space to treat the solver as a negative identity on the backward pass and further provide a theoretical justification. Our experiments demonstrate that such a straightforward hyperparameter-free approach is able to compete with previous more complex methods on numerous experiments such as backpropagation through discrete samplers, deep graph matching, and image retrieval. Furthermore, we substitute the previously proposed problem-specific and label-dependent margin with a generic regularization procedure that prevents cost collapse and increases robustness. Code is available at github.com/martius-lab/solver-differentiation-identity.

## 1 INTRODUCTION

Deep neural networks have achieved astonishing results in solving problems on raw inputs. However, in key domains such as planning or reasoning, deep networks need to make discrete decisions, which can be naturally formulated via constrained combinatorial optimization problems. In many settings—including shortest path finding (Vlastelica et al., 2020; Berthet et al., 2020), optimizing rank-based objective functions (Rolínek et al., 2020a), keypoint matching (Rolínek et al., 2020b; Paulus et al., 2021), Sudoku solving (Amos and Kolter, 2017; Wang et al., 2019), solving the knapsack problem from sentence descriptions (Paulus et al., 2021)—neural models that embed optimization modules as part of their layers achieve improved performance, data-efficiency, and generalization (Vlastelica et al., 2020; Amos and Kolter, 2017; Ferber et al., 2020; P. et al., 2021).

This paper explores the end-to-end training of deep neural network models with embedded discrete combinatorial algorithms (*solvers*, for short) and derives simple and efficient gradient estimators for these architectures. Deriving an informative gradient through the solver constitutes the main challenge, since the true gradient is, due to the discreteness, zero almost everywhere. Most notably, Blackbox Backpropagation (BB) by Vlastelica et al. (2020) introduces a simple method that yields an informative gradient by applying an informed perturbation to the solver input and calling the solver one additional time. This results in a gradient of an implicit piecewise-linear loss interpolation, whose locality is controlled by a hyperparameter.

---

[*]These authors contributed equally

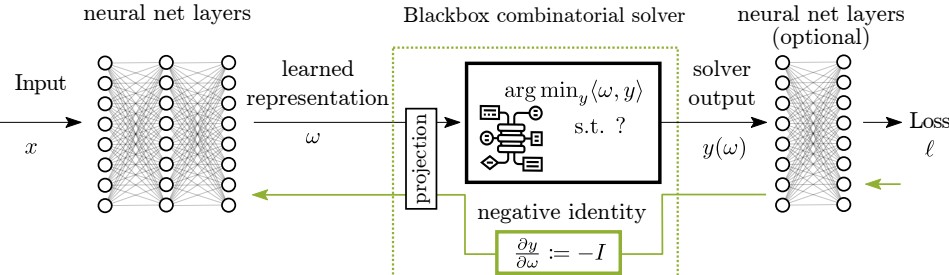

Figure 1: Hybrid architecture with blackbox combinatorial solver and Identity module (green dotted line) with the projection of a cost $\omega$ and negative identity on the backward pass.

We propose a fundamentally different strategy by dropping the constraints on the solver solutions and simply propagating the incoming gradient through the solver, effectively treating the discrete block as a negative identity on the backward pass. While our gradient replacement is simple and cheap to compute, it comes with important considerations, as its naïve application can result in unstable learning behavior, as described in the following.

Our considerations are focused on invariances of typical combinatorial problems under specific transformations of the cost vector. These transformations usually manifest as *projections* or *normalizations*, e.g. as an immediate consequence of the linearity of the objective, the combinatorial solver is agnostic to normalization of the cost vector. Such invariances, if unattended, can hinder fast convergence due to the noise of spurious irrelevant updates, or can result in divergence and cost collapse (Rolínek et al., 2020a). We propose to exploit the knowledge of such invariances by including the respective transformations in the computation graph. On the forward pass this leaves the solution unchanged, but on the backward pass removes the malicious part of the update. We also provide an intuitive view on this as differentiating through a relaxation of the solver. In our experiments, we show that this technique is crucial to the success of our proposed method.

In addition, we improve the robustness of our method by adding noise to the cost vector, which induces a margin on the learned solutions and thereby subsumes previously proposed ground-truth-informed margins (Rolínek et al., 2020a). With these considerations taken into account, our simple method achieves strong empirical performance. Moreover, it avoids a costly call to the solver on the backward pass and does not introduce additional hyperparameters in contrast to previous methods.

Our contributions can be summarized as follows:

 (i) A hyperparameter-free method for linear-cost solver differentiation that does not require any additional calls to the solver on the backward pass.

 (ii) Exploiting invariances via cost projections tailored to the combinatorial problem.

(iii) Increasing robustness and preventing cost collapse by replacing the previously proposed informed margin with a noise perturbation.

(iv) Analysis of the robustness of differentiation methods to perturbations during training.

## 2 RELATED WORK

**Optimizers as Model Building Blocks.**   It has been shown in various application domains that optimization on prediction is beneficial for model performance and generalization. One such area is meta-learning, where methods backpropagate through multiple steps of gradient descent for few-shot adaptation in a multi-task setting (Finn et al., 2017; Raghu et al., 2020). Along these lines, algorithms that effectively embed more general optimizers into differentiable architectures have been proposed such as convex optimization (Agrawal et al., 2019a; Lee et al., 2019), quadratic programs (Amos and Kolter, 2017), conic optimization layers (Agrawal et al., 2019b), and more.

**Combinatorial Solver Differentiation.**   Many important problems require discrete decisions and hence, using *combinatorial* solvers as layers have sparked research interest (Domke, 2012; Elmachtoub and Grigas, 2022). Methods, such as SPO (Elmachtoub and Grigas, 2022) and MIPaaL Ferber et al. (2020), assume access to true target costs, a scenario we are not considering. Berthet et al. (2020) differentiate through discrete solvers by sample-based smoothing. Blackbox Backpropagation

(BB) by Vlastelica et al. (2020) returns the gradient of an implicitly constructed piecewise-linear interpolation. Modified instances of this approach have been applied in various settings such as ranking (Rolínek et al., 2020a), keypoint matching (Rolínek et al., 2020b), and imitation learning (Vlastelica et al., 2020). I-MLE by Niepert et al. (2021) adds noise to the solver to model a discrete probability distribution and uses the BB update to compute informative gradients. Previous works have also considered adding a regularization to the linear program, including differentiable top-$k$-selection (Amos et al., 2019) and differentiable ranking and sorting (Blondel et al., 2020). Another common approach is to differentiate a softened solver, for instance in (Wilder et al., 2019) or (Wang et al., 2019) for MAXSAT. Finally, Paulus et al. (2021) extend approaches for learning the cost coefficients to learning also the constraints of integer linear programs.

**Learning to Solve Combinatorial Problems.**    An orthogonal line of work to ours is differentiable learning of combinatorial algorithms or their improvement by data-driven methods. Examples of such algorithms include learning branching strategies for MIPs (Balcan et al., 2018; Khalil et al., 2016; Alvarez et al., 2017), learning to solve SMT formulas (Balunovic et al., 2018), and learning to solve linear programs (Mandi and Guns, 2020; Tan et al., 2020). A natural way of dealing with the lack of gradient information in combinatorial problems is reinforcement learning which is prevalent among these methods (Khalil et al., 2016; Bello et al., 2017; Nazari et al., 2018; Zhang and Dietterich, 2000). Further progress has been made in applying graph neural networks for learning classical programming algorithms (Velickovic et al., 2018; 2020) and latent value iteration (Deac et al., 2020). Further work in this direction can be found in the review Li et al. (2022). Another interesting related area of work is program synthesis, or "learning to program" (Ellis et al., 2018; Inala et al., 2020).

**Straight-through Estimator (STE).**    The STE (Bengio et al., 2013; Hinton, 2012; Rosenblatt, 1958) differentiates the thresholding function by treating it as an identity function. Jang et al. (2017) extended the STE to differentiating through samples from parametrized discrete distributions. The term STE has also been used more loosely to describe the replacement of any computational block with an identity function (Bengio et al., 2013) or other differentiable replacements (Yin et al., 2019) on the backward pass. Our proposed method can be seen as a generalization of differentiating through the thresholding function to general $\arg\max/\arg\min$ problems with linear objectives, and it is therefore closely connected to the STE. We give a detailed discussion of the relationship in Suppl. A.

## 3   METHOD

We consider architectures that contain differentiable blocks, such as neural network layers, and combinatorial blocks, as sketched in Fig. 1. In this work, a combinatorial block uses an algorithm (called *solver*) to solve an optimization problem of the form

$$y(\omega) = \arg\min_{y \in Y}\langle\omega, y\rangle, \tag{1}$$

where $\omega \in W \subseteq \mathbb{R}^n$ is the cost vector produced by a previous block, $Y \subset \mathbb{R}^n$ is *any finite* set of possible solutions and $y(\omega) \in Y$ is the solver's output. Without loss of generality, $Y$ consists only of extremal points of its convex hull, as no other point can be a solution of optimization (1). This formulation covers linear programs as well as integer linear programs.

### 3.1   DIFFERENTIATING THROUGH COMBINATORIAL SOLVERS

We consider the case in which the solver is embedded inside the neural network, meaning that the costs $\omega$ are predicted by a backbone, the solver is called, and *the solution $y(\omega)$ is post-processed* before the final loss $\ell$ is computed. For instance, this is the case when a specific choice of loss is crucial, or the solver is followed by additional learnable components.

We aim to train the entire architecture end-to-end, which requires computing gradients in a layer-wise manner during backpropagation. However, the true derivative of the solver $y(\omega)$ is *either zero or undefined*, as the relation between the optimal solution $y(\omega)$ and the cost vector $\omega$ is piecewise constant. Thus, it is crucial to contrive a *meaningful replacement* for the true zero Jacobian of the combinatorial block. See Fig. 1 for an illustration.

Note, that the linearity of the objective in (1) makes differentiating the optimization problem more challenging than the non-linear counterparts considered in Amos and Kolter (2017); Agrawal et al. (2019a), due to the inherent discreteness of $Y$. See Suppl. C.1 for a discussion.

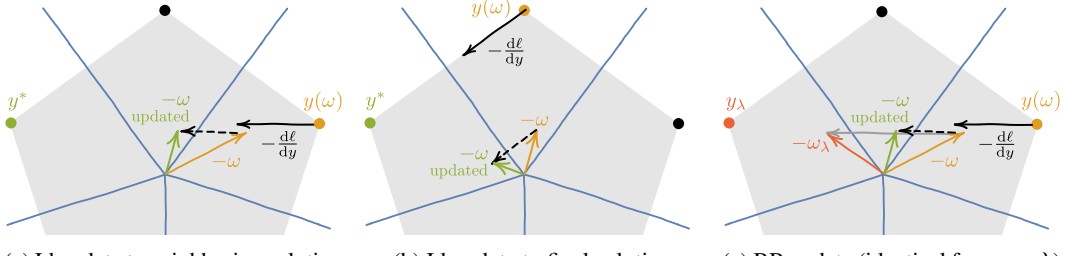

(a) Id update to neighboring solution    (b) Id update to final solution    (c) BB update (identical for some $\lambda$)

Figure 2: Intuitive illustration of the Identity (Id) gradient and its equivalence to Blackbox Backpropagation (BB) when $-\mathrm{d}\ell/\mathrm{d}y$ points directly to a target $y^*$. The cost and solution spaces are overlayed; the cost space partitions resulting in the same solution are drawn in blue. Note that the drawn updates to $\omega$ are only of illustrative nature, as the updates are typically applied to the weights of a backbone.

## 3.2 IDENTITY UPDATE: INTUITION IN SIMPLE CASE

On the backward pass, the negated incoming gradient $-\mathrm{d}\ell/\mathrm{d}y$ gives us the local information of where we expect solver solutions with a lower loss. We first consider the simple scenario in which $-\mathrm{d}\ell/\mathrm{d}y$ points directly toward another solution $y^*$, referred to as the "target". This means that there is some $\eta > 0$ such that $-\eta\frac{\mathrm{d}\ell}{\mathrm{d}y} = y^* - y(\omega)$. This happens, for instance, if the final layer coincides with the $\ell_2$ loss (then $\eta = 1/2$), or for $\ell_1$ loss with $Y$ being a subset of $\{0, 1\}^n$ (then $\eta = 1$).

Our aim is to update $\omega$ in a manner which decreases the objective value associated with $y^*$, i.e. $\langle \omega, y^* \rangle$, and increases the objective value associated with the current solution $y(\omega)$, i.e. $\langle \omega, y(\omega) \rangle$. Therefore, $y^*$ will be favoured over $y(\omega)$ as the solution in the updated optimization problem. This motivates us to set the replacement for the true zero gradient $\mathrm{d}\ell/\mathrm{d}\omega$ to

$$\frac{\mathrm{d}}{\mathrm{d}\omega}\langle \omega, y^* - y(\omega)\rangle = y^* - y(\omega) = -\eta\frac{\mathrm{d}\ell}{\mathrm{d}y}. \tag{2}$$

The scaling factor $\eta$ is subsumed into the learning rate, therefore we propose the update

$$\Delta^{\mathrm{I}}\omega = -\frac{\mathrm{d}\ell}{\mathrm{d}y}. \tag{3}$$

This corresponds to simply treating the solver as a negated identity on the backward pass, hence we call our method "Identity". An illustration of how the repeated application of this update leads to the correct solution is provided in Figures 2a and 2b.

**Comparison to Blackbox Backpropagation.** To strengthen the intuition of why update (3) results in a sensible update, we offer a comparison to BB by Vlastelica et al. (2020) that proposes an update

$$\Delta^{\mathrm{BB}}\omega = \frac{1}{\lambda}\big(y_\lambda(\omega) - y(\omega)\big), \tag{4}$$

where $y(\omega)$ is the solution from the forward pass and, on the backward pass, the solver is called again on a perturbed cost to get $y_\lambda(\omega) = y(\omega + \lambda\mathrm{d}\ell/\mathrm{d}y)$. Here, $\lambda > 0$ is a hyperparameter that controls the locality of the implicit linear interpolation of the loss landscape.

Observe that $y_\lambda(\omega)$ serves as a target for our current $y(\omega)$, similar to the role of $y^*$ in computation (2). If $\lambda$ in BB coincides with a steps size that Identity needs to reach the target $y^*$ with update $\Delta^{\mathrm{I}}\omega$, then

$$\Delta^{\mathrm{BB}}\omega = \frac{1}{\lambda}\Big(y\big(\omega + \lambda\tfrac{\mathrm{d}\ell}{\mathrm{d}y}\big) - y(\omega)\Big) = \frac{1}{\lambda}\big(y^* - y(\omega)\big) = -\frac{\eta}{\lambda}\frac{\mathrm{d}\ell}{\mathrm{d}y} = \frac{\eta}{\lambda}\Delta^{\mathrm{I}}\omega. \tag{5}$$

Therefore, if $-\mathrm{d}\ell/\mathrm{d}y$ points directly toward a neighboring solution, the Identity update is equivalent to the BB update with the smallest $\lambda$ that results in a non-zero gradient in (4). This situation is illustrated in Fig. 2c (cf. Fig. 2a). However, Identity does not require an additional call to the solver on the backward pass, nor does it have an additional hyperparameter that needs to be tuned.

## 3.3 IDENTITY UPDATE: GENERAL CASE

We will now consider the general case, in which we do not expect the solver to find a better solution in the direction $\Delta^{\mathrm{I}}\omega = -\mathrm{d}\ell/\mathrm{d}y$ due to the constraints on $Y$. We show that if we ignore the constraints on $Y$ and simply apply the Identity method, we still achieve a sensible update.

Let $\omega = \omega_0$ be a fixed initial cost. For a fixed step size $\alpha > 0$, we iteratively update the cost using Identity, i.e. we set $\omega_{k+1} = \omega_k - \alpha \Delta^{\mathsf{I}} \omega_k$ for $k \in \mathbb{N}$. We aim to show that performing these updates leads to a solution with a lower loss. As gradient-based methods cannot distinguish between a nonlinear loss $\ell$ and its linearization $f(y) = \ell(y(\omega)) + \langle y - y(\omega), \mathrm{d}\ell/\mathrm{d}y \rangle$ at the point $y(\omega)$, we can safely work with $f$ in our considerations. We desire to find solutions with lower linearized loss than our current solution $y(\omega)$, i.e. points in the set

$$Y^*\big(y(\omega)\big) = \big\{ y \in Y : f(y) < f\big(y(\omega)\big) \big\}. \tag{6}$$

Our result guarantees that a solution with a lower linearized loss is always found if one exists. The proof is in Suppl. C.

**Theorem 1.** *For sufficiently small $\alpha > 0$, either $Y^*\big(y(\omega)\big)$ is empty and $y(\omega_k) = y(\omega)$ for every $k \in \mathbb{N}$, or there is $n \in \mathbb{N}$ such that $y(\omega_n) \in Y^*\big(y(\omega)\big)$ and $y(\omega_k) = y(\omega)$ for all $k < n$.*

### 3.4 Exploiting Solver Invariants

In practice, Identity can lead to problematic cost updates when the optimization problem is invariant to a certain transformation of the cost vector $\omega$. For instance, adding the same constant to every component of $\omega$ will not affect its rank or top-$k$ indices. Formally, this means that there exists a mapping $P\colon \mathbb{R}^n \to \mathbb{R}^n$ of the cost vector $\omega$ that does not change the optimal solver solution, i.e.

$$\arg\min_{y \in Y} \langle \omega, y \rangle = \arg\min_{y \in Y} \langle P(\omega), y \rangle \quad \text{for every } \omega \in W. \tag{7}$$

**Linear Transforms.** Let us demonstrate why invariants can be problematic in a simplified case assuming that $P$ is linear. Consider an incoming gradient $\mathrm{d}\ell/\mathrm{d}y$, for which the Identity method suggests the cost update $\Delta^{\mathsf{I}}\omega = -\mathrm{d}\ell/\mathrm{d}y$. We can uniquely decompose $\Delta^{\mathsf{I}}\omega$ into $\Delta^{\mathsf{I}}\omega = \Delta^{\mathsf{I}}\omega_1 + \Delta^{\mathsf{I}}\omega_0$ where $\Delta^{\mathsf{I}}\omega_1 = P(\Delta^{\mathsf{I}}\omega) \in \operatorname{Im} P$ and $\Delta^{\mathsf{I}}\omega_0 = (I - P)\Delta^{\mathsf{I}}\omega \in \ker P$. Now observe that only the parallel update $\Delta^{\mathsf{I}}\omega_1$ affects the updated optimization problem, as

$$\begin{aligned}
\arg\min_{y \in Y} \langle \omega - \alpha \Delta^{\mathsf{I}}\omega, y \rangle &= \arg\min_{y \in Y} \langle P(\omega - \alpha \Delta^{\mathsf{I}}\omega), y \rangle \\
&= \arg\min_{y \in Y} \langle P(\omega - \alpha \Delta^{\mathsf{I}}\omega_1), y \rangle = \arg\min_{y \in Y} \langle \omega - \alpha \Delta^{\mathsf{I}}\omega_1, y \rangle
\end{aligned} \tag{8}$$

for every $\omega \in W$ and for any step size $\alpha > 0$. In the second equality, we used

$$P(\omega - \alpha \Delta^{\mathsf{I}}\omega) = P(\omega - \alpha \Delta^{\mathsf{I}}\omega_1) - \alpha P(\Delta^{\mathsf{I}}\omega_0) = P(\omega - \alpha \Delta^{\mathsf{I}}\omega_1), \tag{9}$$

exploiting linearity and idempotency of $P$.

In the case when a user has no control about the incoming gradient $\Delta^{\mathsf{I}}\omega = -\mathrm{d}\ell/\mathrm{d}y$, the update $\Delta^{\mathsf{I}}\omega_0$ in $\ker P$ can be much larger in magnitude than $\Delta^{\mathsf{I}}\omega_1$ in $\operatorname{Im} P$. In theory, if updates were applied directly in cost space, this would not be very problematic, as updates in $\ker P$ do not affect the optimization problem. However, in practice, the gradient is further backpropagated to update the weights in the components before the solver. Spurious irrelevant components in the gradient can therefore easily overshadow the relevant part of the update, which is especially problematic as the gradient is computed from stochastic mini-batch samples.

Therefore, it is desirable to discard the irrelevant part of the update. Consequently, for a given incoming gradient $\mathrm{d}\ell/\mathrm{d}y$, we remove the irrelevant part ($\ker P$) and return only its projected part

$$\Delta^{\mathsf{I}}\omega_1 = -P \frac{\mathrm{d}\ell}{\mathrm{d}y}. \tag{10}$$

**Nonlinear Transforms.** For general $P$, we use the chain rule to differentiate the composed function $y \circ P$ and set the solver's Jacobian to identity. Therefore, for a given $\omega$ and an incoming gradient $\mathrm{d}\ell/\mathrm{d}y$, we return

$$-P'(\omega) \frac{\mathrm{d}\ell}{\mathrm{d}y}. \tag{11}$$

If $P$ is linear, then $P'(\omega) = P$ and hence update (11) is consistent with linear case (10). Intuitively, if we replace $P(\omega - \alpha \Delta^{\mathsf{I}}\omega)$ in the above-mentioned considerations by its affine approximation $P(\omega) - \alpha P'(\omega)\Delta^{\mathsf{I}}\omega$, the term $P'(\omega)$ plays locally the role of the linear projection.

Another view on invariant transforms is the following. Consider our combinatorial layer as a composition of two sublayers. First, given $\omega$, we simply perform the map $P(\omega)$ and then pass it to

the $\arg\min$ solver. Clearly, on the forward pass the transform $P$ does not affect the solution $y(\omega)$. However, the derivative of the combinatorial layer is the composition of the derivatives of its sublayers, i.e. $P'(\omega)$ for the transform and negative identity for the $\arg\min$. Consequently, we get the very same gradient (11) on the backward pass. In conclusion, enforcing guarantees on the forward pass by a mapping $P$ is in this sense dual to projecting gradients onto $\mathrm{Im}\,P'$.

**Examples.** In our experiments, we encounter two types of invariant mappings. The first one is the standard projection onto a hyperplane. It is always applicable when all the solutions in $Y$ are contained in a hyperplane, i.e. there exists a unit vector $a \in \mathbb{R}^n$ and a scalar $b \in \mathbb{R}$ such that $\langle a, y \rangle = b$ for all $y \in Y$. Consider the projection of $\omega$ onto the subspace orthogonal to $a$ given by $P_{\mathrm{plane}}(\omega|a) = \omega - \langle a, \omega \rangle a$. This results in

$$\arg\min_{y \in Y} \langle P_{\mathrm{plane}}(\omega|a), y \rangle = \arg\min_{y \in Y} \langle \omega, y \rangle - \langle a, \omega \rangle b = \arg\min_{y \in Y} \langle \omega, y \rangle \quad \text{for every } \omega \in W, \quad (12)$$

thereby fulfilling assumption (7). This projection is relevant for the ranking and top-$k$ experiment, in which the solutions live on a hyperplane with the normal vector $a = \mathbf{1}/\sqrt{n}$. Therefore, the projection

$$P_{\mathrm{mean}}(\omega) = P_{\mathrm{plane}}(\omega|\mathbf{1}/\sqrt{n}) = \omega - \langle \mathbf{1}, \omega \rangle \mathbf{1}/n \quad (13)$$

is applied on the forward pass which simply amounts to subtracting the mean from the cost vector.

The other invariant mapping arises from the stability of the $\arg\min$ solution to the magnitude of the cost vector. Due to this invariance, the projection onto the unit sphere $P_{\mathrm{norm}}(\omega) = \omega/\|\omega\|$ also fulfills assumption (7). As the invariance to the cost magnitude is independent of the solutions $Y$, normalization $P_{\mathrm{norm}}$ is always applicable and we, therefore, test it in every experiment. Observe that

$$P'_{\mathrm{norm}}(\omega) = \left( \frac{I}{\|\omega\|} - \frac{\omega \otimes \omega}{\|\omega\|^3} \right) \quad (14)$$

and the first order approximation of $P_{\mathrm{norm}}$ corresponds to the projection onto the tangent hyperplane given by $a = \omega$ and $b = 1$. When both $P_{\mathrm{norm}}$ and $P_{\mathrm{mean}}$ are applicable, we speak about standardization $P_{\mathrm{std}} = P_{\mathrm{norm}} \circ P_{\mathrm{mean}}$. An illustration of how differentiating through projections affects the resulting gradients is provided in Fig. 9 in Suppl. D. An experimental comparison of the incoming gradient $-\mathrm{d}\ell/\mathrm{d}y$ and the Identity update $\Delta^{\mathrm{I}}\omega_1 = -P'\mathrm{d}\ell/\mathrm{d}y$ is examined in Suppl. B.1.

### 3.5 IDENTITY METHOD AS SOLVER RELAXATION

The Identity method can also be viewed as differentiating a continuous relaxation of the combinatorial solver on the backward pass. To see this connection, we relax the solver by a) relaxing the polytope $Y$ to another smooth set $\widetilde{Y}$ containing $Y$ and b) adding a quadratic regularizer in case $\widetilde{Y}$ is unbounded:

$$y(\omega) = \arg\min_{y \in Y \subset \widetilde{Y}} \langle \omega, y \rangle \quad \rightarrow \quad \widetilde{y}(\omega, \epsilon) = \arg\min_{y \in \widetilde{Y}} \langle \omega, y \rangle + \tfrac{\epsilon}{2}\|y\|^2. \quad (15)$$

Using this viewpoint, we can interpret the vanilla Identity method as the loosest possible relaxation $\widetilde{Y} = \mathbb{R}^n$, and the projections $P_{\mathrm{mean}}$, $P_{\mathrm{norm}}$, and $P_{\mathrm{std}}$ as relaxations of $Y$ to a hyperplane, a sphere, and an intersection thereof, respectively. For a more detailed discussion, see Suppl. C.2.

### 3.6 PREVENTING COST COLLAPSE AND INDUCING MARGIN

Any $\arg\min$ solver (1) is a piecewise constant mapping that induces a partitioning of the cost space $W$ into convex cones $W_y = \{\omega \in W : y(\omega) = y\}$, $y \in Y$, on which the solution does not change. The backbone network attempts to suggest a cost $\omega \in W_y$ that leads to a correct solution $y$. However, if the predicted cost $\omega \in W_y$ lies close to the boundary of $W_y$, it is potentially sensitive to small perturbations. The more partition sets $\{W_{y_1}, \ldots, W_{y_k}\}$ meet at a given boundary point $\omega$, the more brittle such a prediction $\omega$ is, since all the solutions $\{y_1, \ldots, y_k\}$ are attainable in any neighbourhood of $\omega$. For example, the zero cost $\omega = 0$ is one of the most prominent points, as it belongs to the boundary of *all* partition sets. However, the origin is not the only problematic point, for example in the ranking problem, every point $\omega = \lambda\mathbf{1}$, $\lambda > 0$, is *equally bad* as the origin.

The goal is to achieve predictions that are *far* from the boundaries of these partition sets. For $Y = \{0, 1\}^n$, Rolínek et al. (2020a) propose modifying the cost to $\omega' = \omega + \frac{\alpha}{2}y^* - \frac{\alpha}{2}(1 - y^*)$ before the solver to induce an *informed margin* $\alpha$. However, this requires access to the ground-truth solution $y^*$.

We propose to instead add a symmetric noise $\xi \sim p(\xi)$ to the predicted cost before feeding it to the solver. Since all the partition sets' boundaries have zero measure (as they are of a lower dimension), this almost surely induces a margin from the boundary of the size $\mathbb{E}[|\xi|]$. Indeed, if the cost is closer to the boundary, the expected outcome will be influenced by the injected noise and incorrect solutions will increase the expected loss giving an incentive to push the cost further away from the boundary. In experiments, the noise is sampled uniformly from $\left\{-\frac{\alpha}{2}, \frac{\alpha}{2}\right\}^n$, where $\alpha > 0$ is a hyperparameter.

In principle, careful design of the projection map $P$ from Sec. 3.4 can also prevent instabilities. This would require that $\text{Im } P$ avoids the boundaries of the partition sets, which is difficult to fully achieve in practice. However, even *reducing* the size of the problematic set by a projection is beneficial. For example, the normalization $P_{\text{norm}}$ avoids brittleness around the origin, and $P_{\text{mean}}$ avoids instabilities around every $\omega = \lambda \mathbf{1}$ for ranking. For the other—less significant, but still problematic—boundaries, the noise injection still works in general without any knowledge about the structure of the solution set.

## 4 EXPERIMENTS

We present multiple experiments that show under which circumstances Identity achieves performance that is on par with or better than competing methods. The experiments include backpropagating through discrete sampling processes, differentiating through a quadratic assignment solver in deep graph matching, optimizing for rank-based metrics in image retrieval, and a synthetic problem including a traveling salesman solver. The runtimes for all experiments are reported in Sec. B.6.

### 4.1 BACKPROPAGATING THROUGH DISCRETE SAMPLERS

The sampling process of distributions of discrete random variables can often be reparameterized approximately as the solution to a noisy $\arg\max$ optimization problem, see e.g. Paulus et al. (2020). Sampling from the discrete probability distribution $y \sim p(y; \omega)$ can then be formulated as

$$y = \arg\max_{y \in Y} \langle \omega + \epsilon, y \rangle \tag{16}$$

with appropriate noise distribution $\epsilon$. Typically a Gumbel distribution is used for $\epsilon$, but in certain situations, other distributions may be more suitable, e.g. Niepert et al. (2021) use a sum-of-gamma noise distribution to model a top-$k$ distribution on the solutions. Therefore, sampling fits our general hybrid architecture as illustrated in Fig. 1.

**Discrete Variational Auto-Encoder (DVAE).** In a DVAE (Rolfe, 2017), the network layers before the *sampling solver* represent the encoder and the layers after the *sampling solver* the decoder. We consider the task of training a DVAE on the MNIST dataset where the encoder maps the input image to a discrete distribution of $k$-hot binary vector of length 20 in the latent space and the decoder reconstructs the image.

The loss is the Negative Evidence Lower Bound (N-ELBO), which is computed as the sum of the reconstruction losses (binary cross-entropy loss on output pixels) and the KL divergence between the marginals of the discrete latent variables and the uniform distribution. We use the same setup as Jang et al. (2017) and Niepert et al. (2021), details can be found in Suppl. B.1. Figure 3 shows a comparison of Identity to I-MLE (Niepert et al., 2021), which uses the BB update. We observe that Identity (with standardization $P_{\text{std}}$) achieves a significantly lower N-ELBO.

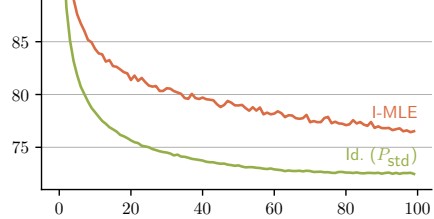

Figure 3: N-ELBO training progress on the MNIST train-set ($k = 10$).

**Learning to Explain.** We consider the BeerAdvocate dataset (McAuley et al., 2012) that consists of reviews of different aspects of beer: appearance, aroma, taste, and palate and their rating scores. The goal is to identify a subset $k$ of the words in the text that best explains a given aspect rating (Chen et al., 2018; Sahoo et al., 2021). We follow the experimental setup of Niepert et al. (2021), details can be found in Suppl. B.1. As in the DVAE experiment, the problem is modeled as a $k$-hot binary latent representation with $k = 5, 10, 15$. Identity also uses the $P_{\text{std}}$.

We compare Identity against I-MLE, L2X (Chen et al., 2018), and Softsub (Xie and Ermon, 2019) in Tab. 1. Softsub is a relaxation-based method designed specifically for subset sampling, in contrast to Identity and I-MLE which are generic. We observe that Identity outperforms I-MLE and L2X, and is on par with Softsub for all $k$. Evaluating Identity without projection shows that projection is indispensable in this experiment.

Table 1: BeerAdvocate (Aroma) statistics over 10 restarts. Projection is indispensable for Identity. Baselines results[†] are from Niepert et al. (2021).

| | Test MSE $\times 100 \downarrow$ | | |
|---|---|---|---|
| Method | $k = 5$ | $k = 10$ | $k = 15$ |
| Id. ($P_{std}$) | $2.62 \pm 0.14$ | $2.47 \pm 0.11$ | $2.39 \pm 0.04$ |
| Id. (no $P$) | $4.75 \pm 0.06$ | $4.43 \pm 0.09$ | $4.20 \pm 0.11$ |
| I-MLE[†] | $2.62 \pm 0.05$ | $2.71 \pm 0.10$ | $2.91 \pm 0.18$ |
| L2X[†] | $5.75 \pm 0.30$ | $6.68 \pm 1.08$ | $7.71 \pm 0.64$ |
| Softsub[†] | $2.57 \pm 0.12$ | $2.67 \pm 0.14$ | $2.52 \pm 0.07$ |

### 4.2 DEEP GRAPH MATCHING

Given a source and a target image showing an object of the same class (e.g. cat), each annotated with a set of keypoints (e.g. right ear), the task is to match the keypoints from visual information without any access to the true keypoint annotation. We follow the standard experimental setup from Rolínek et al. (2020b), in which the cost matrices are predicted by a neural network from the visual information around the keypoints, for details see Suppl. B.2. We report results for the two benchmark datasets Pascal VOC (with Berkeley annotations) (Everingham et al., 2010) and SPair-71k (Min et al., 2019).

Table 2 lists the average matching accuracy across classes. We see that the use of a margin is important for the success of both evaluated methods, which agrees with observations by Rolínek et al. (2020b). We observe that the noise-induced margin achieves a similar effect as the informed margin from Rolínek et al. (2020b). Analogous results for SPair-71k are in the Suppl. B.2.

Our results show that Identity without the projection step yields performance that is on par with BB, while Identity with the normalization step performs worse. Such behaviour is not surprising. Since the Hamming loss between the true and predicted matching is used for training, the incoming gradient for the solver directly points toward the desired solution, as discussed in 3.2. Consequently, any projection step is rendered unnecessary, and possibly even harmful. Interestingly, the evaluation of BB with normalization, a combination of methods that we only evaluated for completeness, appears to produce results that slightly outperform the standard BB.

Table 2: Matching accuracy for Deep Graph Matching on Pascal VOC. Statistics over 5 restarts. ▬ BB ▬ Identity

### 4.3 OPTIMIZING RANK BASED METRICS – IMAGE RETRIEVAL

In image retrieval, the task is to return images from a database that are relevant to a given query. The performance is typically measured with the recall metric, which involves computing the ranking of predicted scores that measure the similarity of image pairs. However, directly optimizing for the recall is challenging, as the ranking operation is non-differentiable.

We follow Rolínek et al. (2020a) in formulating ranking as an $\arg\min$ optimization problem with linear cost, i.e. $\mathbf{rk}(\omega) = \arg\min_{y \in \Pi_n} \langle \omega, y \rangle$, where $\Pi_n$ is the set of all rank permutations and $\omega \in \mathbb{R}^n$ the vector of scores. This allows us to use Identity to directly optimize an architecture for recall. We perform experiments on the image retrieval

Table 3: Recall $R@1$ for CUB-200-2011 with noise-induced margin. ▬ BB ▬ Identity

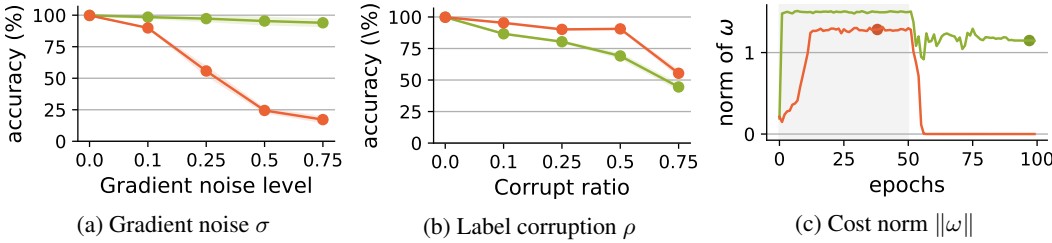

(a) Gradient noise $\sigma$            (b) Label corruption $\rho$            (c) Cost norm $\|\omega\|$

Figure 4: Susceptibility to perturbations and cost collapse in TSP(20). ■■ BB ■■ Identity
(a) Adding noise to the gradient $\mathrm{d}\ell/\mathrm{d}y$ with std $\sigma$ affects Identity much less than BB. (b) Corrupting labels $y^*$ with probability $\rho/k$. (c) Average cost norm with gradient noise $\sigma = 0.25$. The markers indicate the best validation performance.

benchmark CUB-200-2011 (Welinder et al., 2010), and follow the experimental setup used by Rolínek et al. (2020a). Details are provided in Suppl. B.3.

The results are shown in Tab. 3. We report only the noise-induced margin; the informed one performed similarly, see Suppl. B.3. Note that without the standardization $P_{\mathrm{std}}$, Identity exhibits extremely low performance, demonstrating that the projection step is a crucial element of the pipeline. The explanation is that, for ranking-based loss, the incoming gradient does not point toward achievable solutions, as discussed in Sec. 3.2. For more details, including additional evaluation of other applicable projections, see Suppl. B.3. We conclude that for the retrieval experiment, Identity does not match the BB performance. Presumably, this is because the crude approximation of the permutahedron by a sphere ignores too much structural information.

## 4.4 GLOBE TRAVELING SALESMAN PROBLEM

Finally, we consider the Traveling salesman Problem (TSP) experiment from Vlastelica et al. (2020). Given a set of $k$ country flags, the goal for TSP($k$) is to predict the optimal tour on the globe through the corresponding country capitals, without access to the ground-truth location of the capitals. A neural network predicts the location of a set of capitals on the globe, and after computing the pairwise distances of the countries, a TSP solver is used to compute the predicted optimal TSP tour, which is then compared to the ground-truth optimal tour. Setup details can be found in Suppl. B.4.

**Corrupting Gradients.** We study the robustness of Identity and BB to potential perturbations during training in two scenarios: noisy gradients and corrupted labels. We inflict the gradient with noise to simulate additional layers after the solver. In this setting we add $\xi \sim \mathcal{N}(0, \sigma^2)$ to $\mathrm{d}\ell/\mathrm{d}y$. In real-world scenarios, incorrect labels are inevitably occurring. We study random fixed corruptions of labels in the training data, by flipping entries in $y^*$ with probability $\rho/k$.

In Fig. 4a and 4b we observe that Identity performs on par with BB for the standard non-corrupted setting, and outperforms it in the presence of gradient perturbations and label corruptions. Figure 4c shows the average norm of the cost $\omega$ in the course of training under gradient noise with $\sigma = 0.25$. After epoch 50, as soon as $\alpha$ is set to zero, cost collapse occurs quickly for BB, but not for Identity. In all the TSP experiments Identity uses standardization $P_{\mathrm{std}}$.

## 5 CONCLUSION

We present a simple approach for gradient backpropagation through discrete combinatorial solvers with a linear cost function, by treating the solver as a *negative identity* mapping during the backward pass. This approach, in conjunction with the exploitation of invariances of the solver via projections, makes up the *Identity* method. It is hyperparameter-free and does not require any additional computationally expensive call to the solver on the backward pass. We demonstrate in numerous experiments from various application areas that Identity achieves performance that is competitive with more complicated methods and is thereby a viable alternative. We also propose and experimentally verify the use of noise to induce a margin on the solutions and find that noise in conjunction with the projection step effectively increases robustness and prevents cost collapse. Finally, we analyze the robustness of methods to perturbations during training that can come from subsequent layers or incorrect labels and find that Identity is more robust than competing methods.

## ACKNOWLEDGEMENT

This work was supported from Operational Programme Research, Development and Education – Project Postdoc2MUNI (No. CZ.02.2.69/0.0/0.0/18_053/0016952). We thank the International Max Planck Research School for Intelligent Systems (IMPRS-IS) for supporting Anselm Paulus. We acknowledge the support from the German Federal Ministry of Education and Research (BMBF) through the Tübingen AI Center. This work was supported by the Cyber Valley Research Fund.

## REPRODUCIBILITY STATEMENT

We provide extensive experimental details and all used hyperparameters for Discrete Samplers in Suppl. B.1, Deep Graph Matching in Suppl. B.2, Rank Based Metrics in Suppl. B.3, Globe TSP in Suppl. B.4, and Warcraft Shortest Path in Suppl. B.5. The datasets used in the experiments, i.e. BeerAdvocate (McAuley et al., 2012), MNIST (LeCun et al., 2010), SPair-71k (Min et al., 2019), Globe TSP and Warcraft Shortest Path (Vlastelica et al., 2020), CUB-200-2011 (Welinder et al., 2010) are publicly available. A curated Github repository for reproducing all results is available at github.com/martius-lab/solver-differentiation-identity.

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

## A    STRAIGHT THROUGH ESTIMATOR AND IDENTITY

The origins of the Straight-through estimator (STE) are in the perceptron algorithm (Rosenblatt, 1958) for learning single-layer perceptrons. If the output function is a binary variable, the authors use the identity function as a proxy for the zero derivative of the hard thresholding function. Hinton (2012) extended this concept to train multi-layer neural networks with binary activations, and Hubara et al. (2016) used the identity to backpropagate through the sign function, which is known as the saturated STE.

Bengio et al. (2013) considered training neural networks with stochastic binary neurons modelled as random Bernoulli variables $y(p) \in \{0, 1\}$, with $p \in [0, 1]$, which are active (equal to one) with probability $p$ and inactive (equal to zero) with probability $1 - p$. This can be reparametrized using the hard thresholding function as $y(p) = I_{[0,\infty)}(p - \epsilon)$, with $\epsilon \sim \text{Unif}([0, 1])$. In the next step, Bengio et al. (2013) follow previous work by employing the identity as a replacement for the uninformative Jacobian of the hard thresholding function, and coin the term Straight-through estimator for it.

This application of the STE for differentiating through a discrete sampling process allows for an interesting probabilistic interpretation of the STE. To see this, note that the same Jacobian replacement as the one suggested by the STE, i.e. the identity function, is computed by treating the sample $\hat{y}$ on the backward pass as if it were the expected value of the stochastic neuron, i.e.

$$\frac{\mathrm{d}}{\mathrm{d}p}\hat{y}(p) \approx \frac{\mathrm{d}}{\mathrm{d}p}\mathbb{E}\big[y(p)\big] = \frac{\mathrm{d}}{\mathrm{d}p}p = I. \tag{17}$$

This interpretation has also been considered by Niepert et al. (2021); Jang et al. (2017), as it allows the extension of the STE concept to more complicated distributions. Most prominently, it has been applied in the setting of categorical distributions by Jang et al. (2017). Encoding $d$ classes as one hot vectors $Y = \{e_i\}_{i=1}^{d}$, they first define a distribution over classes as a discrete exponential family distribution

$$p_\omega(y) = \frac{\exp(\langle y, \omega \rangle)}{\sum_{\widetilde{y} \in Y} \exp(\langle \widetilde{y}, \omega \rangle)}. \tag{18}$$

Following the STE assumption, we now want to sample from the distribution on the forward pass, and treat it on the backward pass as if it were the expectation of the distribution. We can easily calculate the expectation as

$$\mathbb{E}_{y \sim p_\omega}[y] = \sum_{y \in Y} y \cdot p_\omega(y) = \text{softmax}(\omega), \tag{19}$$

and therefore the generalization of the STE gives the Jacobian replacement

$$\frac{\mathrm{d}}{\mathrm{d}\omega}\hat{y}(\omega) \approx \frac{\mathrm{d}}{\mathrm{d}\omega}\mathbb{E}_{y \sim p_\omega}[y] = \frac{\mathrm{d}}{\mathrm{d}\omega}\text{softmax}(\omega). \tag{20}$$

Jang et al. (2017) arrive at this result through a different reasoning by using the Gumbel-Argmax trick to reparametrize the categorical distribution as a stochastic argmax problem, and then relax the argmax to a softmax (see Paulus et al. (2020) for a more detailed discussion). Due to the derivation involving the Gumbel distribution, they refer to this estimator as the Straight-through Gumbel Estimator.

Note, that for general distributions, the expectation involved in applying an STE according to the previously described scheme is unfortunately intractable to compute, as exponentially many structures need to be considered in the expectation. This is the case even for distributions as simple as the top-$k$ distribution considered in Sec. 4.1. Note, that this is the starting point for the motivation of Niepert et al. (2021), who proceed by applying the concepts observed in Vlastelica et al. (2020) to differentiating through general constrained discrete exponential family distributions.

From the previous examples, we see that our Identity method and the STE are closely related. In fact, treating a hard thresholding function as identity on the backward pass is a special case of our method. To see this, we only need to phrase the thresholding function as the solution to an optimization problem

$$I_{[0,\infty)}(\omega) = \arg\max_{y \in \{0,1\}^d} \langle y, \omega \rangle, \tag{21}$$

for which the Identity method suggests the identity function as a Jacobian replacement, thereby matching the STE. From the discussion above, we know that this can also be interpreted in a probabilistic way for sampling from stochastic Bernoulli processes.

However, it is also apparent that this correspondence does not extend to other distributions. To see this, we reconsider the case of a discrete exponential family distribution over the set of one-hot vectors $Y$. Following Jang et al. (2017) and Maddison et al. (2017), we can simulate the sampling process by computing the solutions to a stochastic argmax problem, i.e.

$$y \sim p_\omega \quad \equiv \quad y = \arg\max_{\widetilde{y} \in Y} \langle \omega + \epsilon, \widetilde{y} \rangle, \quad \epsilon \sim G^d(0,1). \tag{22}$$

Here, $G^d(0,1)$ denotes the $d$-dimensional Gumbel distribution. As described above, applying the STE yields the Jacobian of the softmax function as the replacement for the true zero Jacobian. In contrast, the Identity method (without projections), still returns the identity as the replacement. The benefit of this simplicity is that we can still apply our method to general distributions, where computing the expectation over all structures is intractable.

As a final remark, note that the term STE is somewhat overloaded, as in addition to the formulation above, it has been used to refer to any kind of operation in which some block is treated as an identity function on the backward pass. For example, Bengio et al. (2013) also consider the case of replacing on the backward pass not only the thresholding function, but also a preceding sigmoid computation with the identity function. Yin et al. (2019) even refer to the STE for cases of replacing a thresholding function with a leaky relu function on the backward pass. In this much looser sense, our Identity method also falls into the category of STEs, as we also replace a computational block with the identity function on the backward pass.

## B    EXPERIMENTAL DETAILS

### B.1    BACKPROPAGATING THROUGH DISCRETE SAMPLERS

**DVAE on MNIST.**    Consider the models described by the equations $\theta = f_e(x)$, $y \sim p(y; \theta)$, $z = f_d(y)$ where $x \in \mathcal{X}$ is the input, $o \in \mathcal{O}$ is the output, and $f_e \colon \mathcal{X} \to \theta$ is the encoder neural network that maps the input $x$ to the logits $\theta$ and $f_d \colon \mathcal{Y} \to \mathcal{Z}$ is the decoder neural network, and where $\mathcal{Y}$ is the set of all $k$-hot vectors. Following Niepert et al. (2021), we set $\epsilon$ in sampling (16) as the Sum-of-Gamma distribution given by

$$\mathrm{SoG}(k, \tau, s) = \frac{\tau}{k}\left(\sum_{i=1}^{s} \mathrm{Gamma}\left(\frac{1}{k}, \frac{k}{i}\right) - \log s\right), \tag{23}$$

where $s$ is a positive integer and $\mathrm{Gamma}(\alpha, \beta)$ is the Gamma distribution with $(\alpha, \beta)$ as the shape and scale parameters.

We follow the same training procedure as Niepert et al. (2021). The encoder and the decoder were feedforward neural networks with the architectures: 512-256-20 × 20 and 256-512-784 respectively. We used MNIST (LeCun et al., 2010) dataset for the problem which consisted of $50,000$ training examples and $10,000$ validation and test examples each. We train the model for 100 epochs and record the loss on the test data. In this experiment, we use $k = 10$ i.e. sample 10-hot binary vectors from the latent space. We sample the noise from SoG with $s = 10$ and $\tau = 10$.

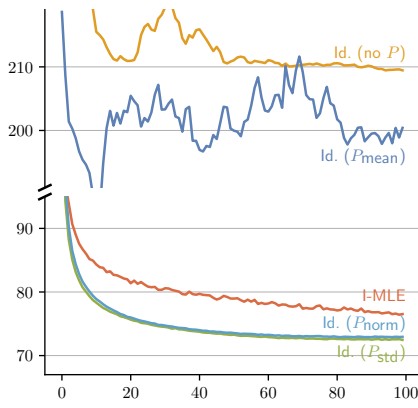

Figure 5: N-ELBO over training epoch for DVAE on MNIST ($k = 10$), comparing I-MLE with Identity for different projections.

**Additional Experimental Results.**    To assess the effect of the individual projections that make up $P_{\mathrm{std}}$, we evaluate our method using only $P_{\mathrm{mean}}$ or $P_{\mathrm{norm}}$. The results reported in Fig. 5 show that both projections increase the performance individually (with a much larger effect of $P_{\mathrm{norm}}$ than $P_{\mathrm{mean}}$), but combined they perform the best.

This is intuitively understandable in terms of viewing projections as solver relaxations (as described in Suppl. C.2). No projection, mean projection, normalization, and standardization correspond to relaxations of $Y$ to $R^n$, a hyperplane $H$, a hypersphere $S$, and the intersection $S \cap H$, respectively, which increasingly better approximate the true $Y$.

Additionally, we examine how the spurious irrelevant part of the update direction returned by the Identity method without projection can lead to problematic behaviour. First, we test whether adaptive optimizers contribute to the problematic behaviour by adapting the learning rate to the irrelevant component of the update. As reported in Fig. 5, we observe a large performance gap between using projections and not using projections in the DVAE experiment. Therefore we rerun this experiment with an SGD optimizer instead of Adam, to check whether a non-adaptive optimizer is able to outperform the adaptive one. The results are reported in Fig. 6, where we observe a slightly improved performance from the SGD optimizer, but no major difference between the the optimizers.

Next, we examine how much differentiating through the projection affects the returned gradient in terms of magnitude and direction. In Fig. 7 we compare the incoming gradient $-\mathrm{d}\ell/\mathrm{d}y$ and the returned update direction $\Delta^{\mathrm{I}}\omega$ after differentiating through the projection in terms of norm ratio and cosine similarity. We observe that the gradient is significantly altered by the projection, i.e. the cosine similarity between the original and the projected gradient is smaller than $0.05$. Given that with the projection the loss is optimized well (in contrast to not using projections), we can conclude that relevant information is indeed retained whereas spurious directions are removed.

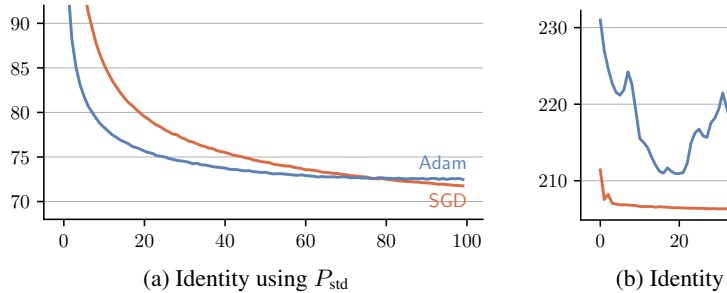

(a) Identity using $P_{\mathrm{std}}$  (b) Identity with no transform $P$.

Figure 6: Training progress of DVAE experiment on the MNIST train-set ($k = 10$), comparing the Identity method for different optimizers. Reported is N-ELBO over training epoch.

**Learning to Explain.** The entire dataset consists of 80k reviews for the aspect Appearance and 70k reviews for all the remaining aspects. Following the experimental setup of Niepert et al. (2021), the dataset was divided into 10 different evenly sized validation/test splits of the 10k held out set. For this experiment, the neural network from Chen et al. (2018) was used which had 4 convolutional and 1 dense layer. The neural network outputs the parameters $\theta$ of the pdf $p(y; \theta)$ over $k$-hot binary latent masks with $k = 5, 10, 15$. The same hyperparameter configuration was the same as that of Niepert et al. (2021). We compute mean and standard deviation over 10 models, each trained on one split.

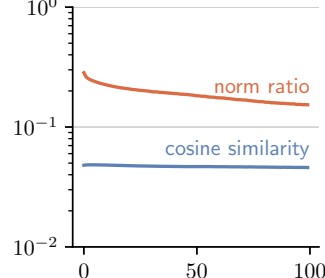

## B.2 DEEP GRAPH MATCHING

We closely follow the experimental setup in Rolínek et al. (2020b), including the architecture of the neural network and the hyperparameter configuration to train the network.

Figure 7: Training progress of DVAE on MNIST ($k = 10$), comparing the incoming gradient $-\mathrm{d}\ell/\mathrm{d}y$ with the update direction computed by the Identity method with projection $P_{\mathrm{std}}$. Reported are the norm ratio and cosine similarity over training epoch.

The architecture consists of a pre-trained backbone that, based on the visual information, predicts feature vectors for each of the keypoints in an image. The keypoints are then treated as nodes of a graph, which is computed via Delauney triangulation. A graph neural network then refines the keypoint features by employing a geometry-aware message passing scheme on the graph. A matching instance is established by computing similarity scores between the edges and nodes of the keypoint graphs of two images via a learnable affinity layer. The negated similarity scores are then used as inputs to

the graph matching solver, which produces a binary vector denoting the predicted matching. This matching is finally compared to the ground-truth matching via the Hamming loss. For a more detailed description of the architecture, see Rolínek et al. (2020b).

We run experiments on SPair-71k (Min et al., 2019) and Pascal VOC (with Berkeley annotations) (Everingham et al., 2010). The Pascal VOC dataset consists of images from 20 classes with up to 23 annotated keypoints. To prepare the matching instances we use intersection filtering of keypoints, i.e. we only consider keypoints that are visible in both images of a pair. For more details on this, we refer the reader to Rolínek et al. (2020b). The SPair-71k dataset consists of $70, 958$ annotated image pairs, with images from the Pascal VOC 2012 and Pascal 3D+ datasets. It comes with a pre-defined train–validation–test split of $53, 340$–$5, 384$–$12, 234$.

We train all models for 10 epochs of 2000 training iterations each, with image pairs processed in batches of 8. As reported in Rolínek et al. (2020b), we set the BB hyperparameter $\lambda = 80$. We use the Adam optimizer with an initial learning rate of $2 \times 10^{-3}$ which is halved every 2 epochs. The learning rate for finetuning the VGG weights is multiplied by $10^{-2}$. As in Rolínek et al. (2020b) we use a state-of-the-art dual block coordinate ascent solver (Swoboda et al., 2017) based on Lagrange decomposition to solve the combinatorial graph matching problem.

We report the results from Tab. 2 again with their corresponding numbers in Tab. 4.

Table 4: Matching accuracy ($\uparrow$) for Deep Graph Matching.

| $P$ | margin | $\alpha$ | Pascal VOC Identity | BB | | SPair-71k Identity | BB | |
|---|---|---|---|---|---|---|---|---|
| | – | – | $77.2 \pm 0.7$ | $74.9 \pm 0.9$ | | $78.6 \pm 0.4$ | $78.1 \pm 0.4$ | |
| | noise | 0.01 | $77.6 \pm 0.7$ | $74.6 \pm 0.8$ | | $78.7 \pm 0.3$ | $77.9 \pm 0.5$ | |
| | | 0.1 | $78.6 \pm 0.4$ | $78.0 \pm 0.9$ | | $79.1 \pm 0.6$ | $78.8 \pm 0.7$ | |
| | | 1.0 | $79.6 \pm 0.8$ | $80.1 \pm 0.4$ | | $79.0 \pm 0.3$ | $78.5 \pm 0.4$ | |
| – | | 10.0 | $78.8 \pm 0.6$ | $79.8 \pm 0.4$ | | $77.7 \pm 0.4$ | $78.2 \pm 0.4$ | |
| | informed | 0.01 | $78.5 \pm 0.5$ | $76.8 \pm 0.6$ | | $78.9 \pm 0.5$ | $78.7 \pm 0.3$ | |
| | | 0.1 | $79.7 \pm 0.6$ | $79.3 \pm 0.6$ | | $79.1 \pm 0.3$ | $78.9 \pm 0.4$ | |
| | | 1.0 | $79.0 \pm 0.6$ | $80.1 \pm 0.6$ | | $78.5 \pm 0.2$ | $78.6 \pm 0.1$ | |
| | | 10.0 | $77.9 \pm 0.5$ | $78.7 \pm 0.6$ | | $77.2 \pm 0.5$ | $68.6 \pm 20$ | |
| | – | – | $72.6 \pm 1.0$ | $63.5 \pm 0.8$ | | $75.9 \pm 0.8$ | $66.7 \pm 0.6$ | |
| | noise | 0.01 | $76.3 \pm 0.1$ | $78.9 \pm 0.5$ | | $77.1 \pm 0.3$ | $78.2 \pm 0.4$ | |
| | | 0.1 | $78.1 \pm 1.1$ | $80.1 \pm 0.4$ | | $77.3 \pm 1.0$ | $78.5 \pm 0.2$ | |
| | | 1.0 | $79.1 \pm 0.5$ | $81.2 \pm 0.5$ | | $77.7 \pm 0.2$ | $80.3 \pm 0.3$ | |
| $P_{\text{norm}}$ | | 10.0 | $79.0 \pm 0.7$ | $81.1 \pm 0.3$ | | $77.6 \pm 0.4$ | $80.3 \pm 0.3$ | |
| | informed | 0.01 | $76.9 \pm 0.7$ | $79.0 \pm 1.0$ | | $77.7 \pm 0.3$ | $78.5 \pm 0.2$ | |
| | | 0.1 | $78.4 \pm 1.1$ | $79.7 \pm 0.9$ | | $77.7 \pm 0.3$ | $78.7 \pm 0.3$ | |
| | | 1.0 | $73.7 \pm 1.0$ | $80.8 \pm 0.4$ | | $74.4 \pm 0.7$ | $80.2 \pm 0.4$ | |
| | | 10.0 | $74.2 \pm 0.9$ | $79.7 \pm 0.6$ | | $74.4 \pm 0.6$ | $80.1 \pm 0.2$ | |

## B.3 RANK-BASED METRICS – IMAGE RETRIEVAL EXPERIMENTS

**Ranking as a Solver.** Ranking can be cast as a combinatorial $\arg\min$ optimization problem with linear cost by a simple application of the permutation inequality as described in Rolínek et al. (2020a). It holds that

$$\mathbf{rk}(\omega) = \arg\min_{y \in \Pi_n} \langle \omega, y \rangle, \tag{24}$$

where $\Pi_n$ is the set of all rank permutations and $\omega \in \mathbb{R}^n$ is the vector of individual scores.

A popular metric in the retrieval literature is recall at $K$, denoted by $r@K$, which can be formulated using ranking as

$$r@K(\omega, y^*) = \begin{cases} 1 & \text{if there is } i \in \text{rel}(y^*) \text{ with } \mathbf{rk}(\omega)_i \leq K \\ 0 & \text{otherwise,} \end{cases} \tag{25}$$

where $K \in \mathbb{N}$, $\omega \in \mathbb{R}^n$ are the scores, $y^* \in \{0, 1\}^n$ are the ground-truth labels and $\text{rel}(y^*) = \{i : y_i^* = 1\}$ is the set of relevant label indices (positive labels). Consequently, $r@K$ is going to be 1 if

a positively labeled element is among the predicted top $K$ ranked elements. Due to computational restrictions, at training time, instead of computing $r@K$ over the whole dataset a sample-based version is used.

**Recall Loss.** Since $r@K$ depends only on the top-ranked element from $y^*$, the supervision for $r@K$ is very sparse. To circumvent this, Rolínek et al. (2020a) propose a loss

$$\ell_{\text{recall}}(\omega, y^*) = \frac{1}{|\operatorname{rel}(y^*)|} \sum_{i \in \operatorname{rel}(y^*)} \log\big(1 + \log\big(1 + \mathbf{rk}(\omega)_i - \mathbf{rk}(\omega^+)_i\big)\big), \tag{26}$$

where $\mathbf{rk}(\omega^+)_i$ denotes the rank of the $i$-th element only within the relevant ones. This loss is called *loglog* in the paper that proposed it. We follow their approach and use this loss in all image retrieval experiments.

**Experimental Configuration.** We closely follow the training setup and procedure reported in Rolínek et al. (2020a). The model consists of a pre-trained ResNet50 (He et al., 2016) backbone in which the final softmax is replaced with a fully connected embedding layer, producing a 512-dimensional embedding vector for each batch element. These vectors are then used to compute pairwise similarities between batch elements. The ranking of these similarities is then used to compute the recall loss (26), for additional details we refer the reader to Rolínek et al. (2020a).

We train all models for 80 epochs using the Adam optimizer, using a learning rate of $5 \times 10^{-7}$, which was the best performing learning rate for both Identity and BB out of a grid search over 5 learning rates. In all experiments a weight decay of $4 \times 10^{-4}$ is used, as well as a drop of the learning rate by 70% after 35 epochs. The learning rate of the embedding layer is multiplied by 3, following previous work. The BB parameter $\lambda$ was set to 0.2 for all retrieval experiments. Images are processed in batches of 128, and three previous batches are always kept in memory to compute the recall to better approximate the recall over the whole dataset, see Rolínek et al. (2020a) for more details.

**Sparse Gradient & Difference between Identity and BB.** As described in Sec. 4.3, Identity performs worse than BB in the retrieval experiments. The reason is the sparsity of the gradient generated from the loss (26): it only yields a nonzero gradient for the relevant images (positive labels), and provides no information for the images irrelevant to the query. This sparseness makes training very inefficient.

The BB update circumvents this by producing a new ranking $y_\lambda$ for a perturbed input. The difference between the two rankings is then returned as the gradient, which contains information for both positive and negative examples. This process is illustrated in Figure 8a.

**Ablations.** To assess the effect of the individual projections that make up $P_{\text{std}}$, we also evaluate our method using only $P_{\text{mean}}$ or $P_{\text{norm}}$ and report them in Tab. 5. We observe the same trend as for the DVAE ablations reported in Fig. B.1, and therefore refer to the same explanation in terms of increasingly tight relaxations of the solution set.

## B.4 GLOBE TSP

**Dataset.** The training dataset for TSP($k$) consists of $10\,000$ examples where each datapoint has a $k$ element subset sampled from 100 country flags as input and the output is the optimal TSP tour represented by an adjacency matrix. We consider datasets for $k = 5, 10, 20$. For additional details, see Vlastelica et al. (2020).

**Architecture.** A set of $k$ flags is presented to a convolutional neural network that outputs $k$ 3-dimensional coordinates. These points are then projected onto a 3D unit sphere and then used to construct the $k \times k$ distance matrix which is fed to a TSP solver. Then the solver outputs the adjacency matrix indicating the edges present in the TSP tour. The loss function is an L1 loss between the predicted tour and the true tour. The neural network is expected to learn the correct coordinates of the countries' capitals on Earth, up to rotations of the sphere. We use the `gurobi` (Gurobi Optimization, LLC, 2022) solver for the MIP formulation of TSP.

For the Globe TSP experiment we use a convolutional neural network with 2 conv. layers ((channels, kernel_size, stride) = [[20, 4, 2], [50, 4, 2]]) and 1 fully connected layer of size 500 that predicts

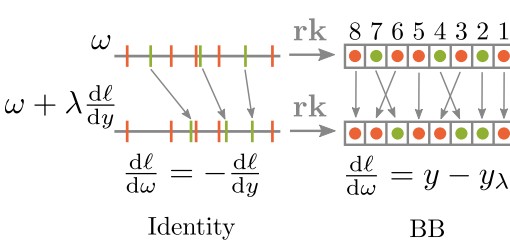

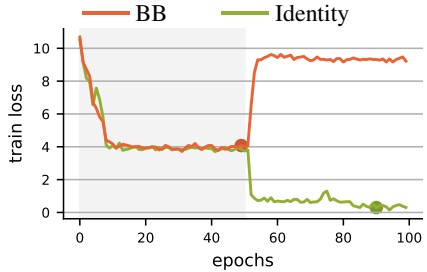

(a) Visualization of the different ranking gradients.

(b) Training for TSP(5) for gradient noise level 0.5.

Figure 8: (a) Left: the scores $\omega$, with $\mathrm{rel}(y^*)$ in green and their shifted counterparts below. The grey arrows correspond to the Identity-gradient. Right: BB performs another ranking $y_\lambda$ with the shifted $\omega$ which yields a denser gradient. (b) To check for the robustness of the methods, we apply gradient noise during the entire training time (100 epochs). The shaded grey region highlights the epochs for which margin-inducing noise ($\xi$) was applied to the inputs to prevent cost collapse. The dots on the training curve represent early stopping epochs.

Table 5: Recall $R@1$ ($\uparrow$) for CUB-200-2011 with various projections $P$ and margin types.

| $P$ | $\alpha$ | Noise-induced margin | | | Informed margin | |
|---|---|---|---|---|---|---|
| | | Identity ▬ | BB ▬ | | Identity ▬ | BB ▬ |
| _ | 0 | $13.9 \pm 0.9$ | $62.6 \pm 0.4$ | | $13.9 \pm 0.9$ | $62.6 \pm 0.4$ |
| | 0.001 | $14.5 \pm 1.2$ | $62.6 \pm 0.3$ | | $16.2 \pm 1.7$ | $63.0 \pm 0.3$ |
| | 0.01 | $13.6 \pm 0.8$ | $62.6 \pm 0.5$ | | $25.5 \pm 1.2$ | $63.0 \pm 0.2$ |
| | 0.1 | $13.4 \pm 1.0$ | $62.0 \pm 0.2$ | | $28.9 \pm 2.7$ | $62.8 \pm 0.4$ |
| $P_{\mathrm{mean}}$ | 0 | $44.6 \pm 0.8$ | $62.9 \pm 0.5$ | | $44.6 \pm 0.8$ | $62.9 \pm 0.5$ |
| | 0.001 | $44.0 \pm 1.0$ | $62.9 \pm 0.2$ | | $44.3 \pm 0.8$ | $62.9 \pm 0.5$ |
| | 0.01 | $44.5 \pm 0.4$ | $62.9 \pm 0.5$ | | $45.9 \pm 1.0$ | $62.7 \pm 0.2$ |
| | 0.1 | $44.9 \pm 0.4$ | $61.9 \pm 0.6$ | | $49.9 \pm 0.5$ | $62.1 \pm 0.3$ |
| $P_{\mathrm{norm}}$ | 0 | $59.0 \pm 0.4$ | $60.8 \pm 0.3$ | | $59.0 \pm 0.4$ | $60.8 \pm 0.3$ |
| | 0.001 | $59.3 \pm 0.5$ | $62.6 \pm 0.5$ | | $58.7 \pm 0.5$ | $62.7 \pm 0.3$ |
| | 0.01 | $58.9 \pm 0.5$ | $62.5 \pm 0.4$ | | $58.3 \pm 0.6$ | $62.3 \pm 0.6$ |
| | 0.1 | $59.3 \pm 0.6$ | $60.1 \pm 0.2$ | | $50.4 \pm 0.5$ | $50.4 \pm 0.4$ |
| $P_{\mathrm{std}}$ | 0 | $60.2 \pm 0.6$ | $62.4 \pm 0.6$ | | $60.2 \pm 0.6$ | $62.4 \pm 0.6$ |
| | 0.001 | $60.0 \pm 0.6$ | $62.7 \pm 0.3$ | | $60.5 \pm 0.1$ | $63.0 \pm 0.7$ |
| | 0.01 | $60.4 \pm 0.5$ | $63.0 \pm 0.5$ | | $59.9 \pm 0.4$ | $63.2 \pm 0.3$ |
| | 0.1 | $60.5 \pm 0.4$ | $62.1 \pm 0.2$ | | $50.3 \pm 1.1$ | $61.3 \pm 0.4$ |

vector of dimension $3k$ containing the $k$ 3-dimensional representations of the respective countries' capital cities. These representations are projected onto the unit sphere and the matrix of pairwise distances is fed to the TSP solver. The network was trained using Adam optimizer with a learning rate $10^{-4}$ for 100 epochs and a batch size of 50. For BB, the hyper-parameter $\lambda$ was set to 20.

As described in Sec. 3.6 adding noise to the cost vector can avoid cost collapse and act as a margin-inducing procedure. We apply noise, $\xi = 0.1, 0.2, 0.5$, for the first 50 epochs (of 100 in total) to prevent cost collapse. For this dataset, it was observed that finetuning the weights after applying noise helped improve the accuracy. Margin is only important initially as it allows not getting stuck in a local optimum around zero cost. Once avoided, margin does not play a useful role anymore because there are no large distribution shifts between the train and test set. Hence, noise was not applied for the entirety of the training phase. We verified the benefits of adding noise experimentally in Tab. 6.

A difference between BB and Identity can be observed when applying gradient noise, simulating larger architectures where the combinatorial layer is followed by further learnable layers. In Fig. 8b, we present the training loss curves in this case. As we see, BB starts to diverge right after the margin-inducing noise is not added anymore i.e. after epoch 50. However, Identity converges as the training progresses.

|  | $\xi = 0$ | $\xi = 0.1$ | $\xi = 0.2$ | $\xi = 0.5$ |
|---|---|---|---|---|
| TSP(5) | $91.09 \pm 0.07$ | $99.67 \pm 0.10$ | $99.67 \pm 0.10$ | $99.68 \pm 0.09$ |
| TSP(10) | $85.31 \pm 0.15$ | $99.72 \pm 0.04$ | $99.76 \pm 0.07$ | $99.76 \pm 0.04$ |
| TSP(20) | $88.45 \pm 0.88$ | $99.78 \pm 0.04$ | $94.50 \pm 10.53$ | $99.80 \pm 0.06$ |

Table 6: Adding noise $\xi$ during the initial 50 epochs of training prevents cost collapse and therefore improves the test accuracy.

## B.5 WARCRAFT SHORTEST PATH

We additionally present an evaluation of the Warcraft Shortest Path experiment from Vlastelica et al. (2020). Here the aim is to predict the shortest path between the top-left and bottom-right vertex in a $k \times k$ Warcraft terrain map. The path is represented by an indicator matrix of vertices that appear along the path.

The non-negative vertex costs of the $k \times k$ grid are computed by a modified Resnet18 (He et al., 2016) architecture using softplus on the output. The network receives supervision in form of L1 loss between predicted and target paths. We consider only the hardest case from the dataset with map sizes $32 \times 32$.

Table 7: Cost ratio (suggested vs. true path costs) for Warcraft Shortest Path $32 \times 32$. BB and Identity work similarly well. Normalization $P_{\text{norm}}$ or noise does not affect the performance significantly.

|  |  | Cost ratio $\times 100 \downarrow$ | |
|---|---|---|---|
| $P$ | $\alpha$ | BB | Identity |
| $-$ | 0 | $100.9 \pm 0.1$ | $101.0 \pm 0.1$ |
| $P_{\text{norm}}$ | 0 | $100.9 \pm 0.1$ | $101.2 \pm 0.1$ |
| $-$ | 0.2 | $101.1 \pm 0.1$ | $101.0 \pm 0.1$ |

Due to the non-uniqueness of solutions, we use the ratio between true and predicted path costs as an evaluation metric: $\langle \omega^*, y(\omega) \rangle / \langle \omega^*, y^* \rangle$, with $\omega^*$ being the ground truth cost vector. Note that lower scores are better and 1.0 is the perfect score. Table 7 shows that Identity performs comparatively to BB.

We follow the same experimental design as Vlastelica et al. (2020) and do the same modification to the ResNet18 architecture, except that we use *softplus* to make the network output positive. The model is trained with Adam for 50 epochs with learning rate $5 \times 10^{-3}$. The learning rate is divided by 10 after 30 epochs. For the BB method we use $\lambda = 20$. The noise, when used, is applied for the whole duration of training.

## B.6 RUNTIMES

We report the runtimes of our method and BB for all experiments in Table 8. Without surprise, we see the largest improvement of Identity over BB in terms of runtime when the underlying combinatorial problem is difficult. Finding the top-$k$ indices or the ranking of a vector is extremely simple, and therefore this procedure is typically not the computational bottleneck of the architecture. Consequently, we do not see any difference between the methods in terms of runtime for the discrete sampling experiments (DVAE and L2X) and the retrieval experiment. In the matching experiment, the matching procedure is makes up for a relevant portion of training, which also observed in the runtime, as Identity requires less training time for the same number of training iterations. The runtime of the quadratic assignment solver scales with the number of keypoints in the image pairs, which is between between 9 and 30 in our experiments, and therefore the difference in runtime between the methods will also increase for instances with more keypoints to match. Finally, the TSP problem poses the hardest of the considered combinatorial problems, which is reflected in the almost halved runtime of the architecture employing Identity compared to using BB.

Overall, we conclude that choosing between Identity and BB depends both on the difficulty of the combinatorial problem, and the performance of the two methods, which is affected by the quality of the incoming gradient as discussed in Sec. 3.3.

|  |  | Sampling | Graph Matching | | Retrieval | TSP |
|--|--|----------|----------------|--|-----------|-----|
|  |  | L2X | Pascal VOC | SPair-71k | CUB-200-2011 | $k = 10$ |
| train | Identity | $38.0 \pm 0.6$ | $97.0 \pm 2.5$ | $93.5 \pm 1.9$ | $99.7 \pm 4.0$ | $53.1 \pm 2.5$ |
|  | BB | $37.6 \pm 1.5$ | $104.4 \pm 2.2$ | $101.7 \pm 2.0$ | $99.5 \pm 3.4$ | $84.6 \pm 0.7$ |
| eval | Identity | $0.3 \pm 0$ | $7.5 \pm 0.3$ | $7.0 \pm 0.2$ | $87.1 \pm 3.8$ | $0.1 \pm 0$ |
|  | BB | $0.3 \pm 0$ | $7.5 \pm 0.3$ | $7.3 \pm 0.2$ | $88.0 \pm 5.5$ | $0.1 \pm 0$ |

Table 8: Runtimes (in minutes) for performed exeeperiments.

## C    METHOD

### C.1    RELATION TO NON-LINEAR IMPLICIT DIFFERENTIATION

In the case of a linear program (1), the optimal solution is always located at one of the corners of the conex hull of the possible solutions, where the Jacobian of the optimal solution with respect to the ocost vector is zero. Note, that this is in contrast to the case of optimization problems with non-linear objectives over convex regions, such as OptNet (Amos and Kolter, 2017) and Cvxpy (Agrawal et al., 2019a). Intuitively, here the non-linearity often leads to solutions that are not located at such problematic corner points, and therefore the optimal solution usually has a continuously differentiable dependence on the objective parameters with non-zero Jacobian. This Jacobian can be calculated by solving the KKT system on the backward pass, and therefore no informative gradient replacement is required, given a sufficiently strong regularization through the non-linearity.

### C.2    PROJECTION AS SOLVER RELAXATION

As described in Sec. 3.5, we can also view the Identity method in terms of relaxations of the combinatorial solver. From this perspective, the approach is to use the unmodified discrete solver on the forward pass but differentiate through a continuous relaxation on the backward pass. To see this connection, we modify the solver by a) relaxing the polytope $Y$ to another smooth set $\widetilde{Y}$ containing $Y$ and b) adding a quadratic regularizer in case $\widetilde{Y}$ is unbounded, i.e.

$$y(\omega) = \arg\min_{y \in Y \subset \widetilde{Y}} \langle \omega, y \rangle \quad \rightarrow \quad \widetilde{y}_{\widetilde{Y}}(\omega, \epsilon) = \arg\min_{y \in \widetilde{Y}} \langle \omega, y \rangle + \frac{\epsilon}{2} \|y\|_2^2. \tag{27}$$

The loosest possible relaxation is to set $\widetilde{Y} = \mathbb{R}^n$. This gives

$$\widetilde{y}_{\mathbb{R}^n}(\omega, \epsilon) = \arg\min_{y \in \mathbb{R}^n} \langle \omega, y \rangle + \frac{\epsilon}{2} \|y\|_2^2 = -\frac{\omega}{\epsilon} \tag{28}$$

as the closed form solution of the relaxed problem, and differentiating through it corresponds to the re-scaled vanilla Identity method. We can improve upon this by making the relaxation tighter. As an example, we can restrict the solution space to a hyperplane $H = \{y \in \mathbb{R}^n | \langle a, y \rangle = b\}$ for some unit vector $a \in \mathbb{R}^n$ and scalar $b \in \mathbb{R}$, and setting $\widetilde{Y} = H$ leads to the closed-form solution

$$\widetilde{y}_H(\omega, \epsilon) = \arg\min_{y \in H} \langle \omega, y \rangle + \frac{\epsilon}{2} \|y\|_2^2 = -\frac{1}{\epsilon}(\omega - \langle \omega, a \rangle a) + b = -\frac{1}{\epsilon} P_{\text{plane}}(\omega|a) + b. \tag{29}$$

The Jacobian of this expression matches that of Identity with $P_{\text{plane}}$ up to scale. In practice, we also often encounter the case in which $Y$ is a subset of a sphere $S = c + r\mathbb{S}_{n-1}$, for some $c \in \mathbb{R}^n, r > 0$. This is the case in all of our experiments, as both the binary hypercube and the permutahedron are subsets of certain spheres. Specifically, that is

$$c = \frac{\sqrt{n}}{2}\mathbf{1}, \qquad r = \frac{\sqrt{n}}{2} \tag{30}$$

for the binary hypercube, and

$$c = \frac{n+1}{2}\mathbf{1}, \qquad r = \sqrt{\frac{n(n^2-1)}{12}} \tag{31}$$

for the permutahedron. Setting $\widetilde{Y} = S$ gives the closed form solution

$$\widetilde{y}_S(\omega, \epsilon = 0) = \arg\min_{y \in S} \langle \omega, y \rangle = c - r\frac{\omega}{\|\omega\|} = c - r P_{\text{norm}}(\omega). \tag{32}$$

The Jacobian of this relaxed solution is proportional to that of Identity with projection $P_{\text{norm}}$. In our experiments, we also encounter the case in which the solutions are located on the intersection of a hyperplane $H$ and a sphere $S$. We can therefore tighten the relaxation further by $\widetilde{Y} = S \cap H$.

Assuming without loss of generality that the sphere center is located on the hyperplane, we get the closed form solution

$$\widetilde{y}_{S \cap H}(\omega, \epsilon = 0) = \arg\min_{y \in S \cap H} \langle \omega, y \rangle = c - r\frac{\omega - \langle \omega, a \rangle a}{\|\omega - \langle \omega, a \rangle a\|} = c - r P_{\text{norm}}(P_{\text{plane}}(\omega|a)). \tag{33}$$

In the case of $Y$ being the set of $k$-hot vectors or the permutahedron, we have $a = 1/\sqrt{n}$, which amounts to

$$\widetilde{y}_{S \cap H}(\omega, \epsilon = 0) = c - r P_{\text{norm}}(P_{\text{plane}}(\omega|a = \frac{1}{\sqrt{n}})) = c - r P_{\text{std}}(\omega). \tag{34}$$

The Jacobian of this expression matches that of Identity using $P_{\text{std}}$ projection, which we derived originally by considering solver invariants in Sec. 3.4. This further strengthens the intuitive connection between projections in cost-space and relaxations in solution-space.

This view of Identity is also related to existing work on structured prediction with projection oracles (Blondel, 2019), which uses (potentially non-differentiable) projections onto convex super-sets on both the forward- and backwards pass.

### C.3   PROOF OF THEOREM 1

For an initial cost $\omega$ and a step size $\alpha > 0$, we set

$$\omega_0 = \omega \quad \text{and} \quad \omega_{k+1} = \omega_k - \alpha \Delta^{\text{I}} \omega_k \quad \text{for } k \in \mathbb{N}, \tag{35}$$

in which $\Delta^{\text{I}} \omega_k$ denotes the Identity update at the solution point $y(\omega_k)$, i.e.

$$\Delta^{\text{I}} \omega_k = -\frac{\text{d}\ell}{\text{d}y}\big(y(\omega_k)\big). \tag{36}$$

We shall simply write $\text{d}\ell/\text{d}y$ when no confusion is likely to happen. Recall that the set of better solutions is defined as

$$Y^*(y) = \{y' \in Y : f(y') < f(y)\}, \tag{37}$$

where $f$ is the linearization of the loss $\ell$ at the point $y$ defined by

$$f(y') = \ell(y) + \left\langle y' - y, \frac{\text{d}\ell}{\text{d}y} \right\rangle. \tag{38}$$

In principle, ties in the solver may occur and hence the mapping $\omega \mapsto y(\omega)$ is not well-defined for all cost $\omega$ unless we specify, how the ties are resolved. Typically, this is not an issue in most of the considerations. However, in our exposition, we need to avoid certain rare discrepancies. Therefore, we assume that the solver will always favour the solution from the previous iteration if possible, i.e. from $\langle y(\omega_k), \omega_k \rangle = \langle y(\omega_{k-1}), \omega_k \rangle$ it follows that $y(\omega_k) = y(\omega_{k-1})$.

**Theorem 2.** *Assume that $(\omega_k)_{k=0}^{\infty}$ is the sequence as in (35) for some initial cost $\omega$ and step size $\alpha > 0$. Then the following holds:*

   (i) *Either $Y^*\big(y(\omega)\big)$ is non-empty and there is some $\alpha_{\max} > 0$ such that for every $\alpha < \alpha_{\max}$ there is $n \in \mathbb{N}$ such that $y(\omega_n) \in Y^*\big(y(\omega)\big)$ and $y(\omega_k) = y(\omega)$ for all $k < n$,*

  (ii) *or $Y^*\big(y(\omega)\big)$ is empty and for every $\alpha$ it is $y(\omega_k) = y(\omega)$ for all $k \in \mathbb{N}$.*

We prove this statement in multiple parts.

**Proposition 1.** *Let $\alpha > 0$ and $(\omega_k)_{k=0}^{\infty}$ be as in (35). If $Y^*\big(y(\omega)\big)$ is non-empty, then there exists $n \in \mathbb{N}$ such that $y(\omega_n) \neq y(\omega)$.*

*Proof.* We proceed by contradiction. Assume that $Y^*\big(y(\omega)\big)$ is non-empty and $y(\omega_k) = y(\omega)$ for all $k \in \mathbb{N}$.

Take any $y^* \in Y^*\big(y(\omega)\big)$. By definition of $Y^*\big(y(\omega)\big)$, we have that

$$\xi = \Big\langle y^* - y(\omega), \frac{\mathrm{d}\ell}{\mathrm{d}y} \Big\rangle < 0. \tag{39}$$

As $y(\omega_k) = y(\omega)$ for all $k \in \mathbb{N}$, it is

$$\omega_k = \omega - k\alpha\Delta^{\mathrm{l}}\omega = \omega + k\alpha\frac{\mathrm{d}\ell}{\mathrm{d}y} \tag{40}$$

and therefore

$$\langle y^* - y(\omega), \omega_k \rangle = \langle y^* - y(\omega), \omega \rangle + k\alpha\Big\langle y^* - y(\omega), \frac{\mathrm{d}\ell}{\mathrm{d}y} \Big\rangle = \langle y^* - y(\omega), \omega \rangle + k\alpha\xi. \tag{41}$$

Since $\xi < 0$, the latter term tends to minus infinity. Consequently, there exists some $n \in \mathbb{N}$ for which

$$\langle y^*, \omega_n \rangle < \langle y(\omega), \omega_n \rangle \tag{42}$$

contradicting the fact that $y(\omega)$ is the minimizer for $\omega_n$. □

Let us now make a simple auxiliary observation about $\arg\min$ solvers. The mapping

$$\omega \mapsto y(\omega) = \arg\min_{y \in Y}\langle\omega, y\rangle \tag{43}$$

is a piecewise constant function and hence induces a partitioning of its domain $W$ into non-overlapping sets on which the solver is constant. Let us denote the pieces by

$$W_y = \{\omega \in W : y(\omega) = y\} \quad \text{for } y \in Y. \tag{44}$$

We claim that $W_y$ is a convex cone. Indeed, if $\omega \in W_y$, clearly $\lambda\omega \in W_y$ for any $\lambda > 0$. Next, if $\omega_1, \omega_2 \in W_y$ and $\lambda \in (0,1)$, then $y(\lambda\omega_1) = y$ and $y\big((1-\lambda)\omega_2\big) = y$ and hence $y$ is also a minimizer for $\lambda\omega_1 + (1-\lambda)\omega_2$.

**Proposition 2.** *Assume that $(\omega_k)_{k=0}^\infty$ is the sequence as in* (35) *for some initial cost $\omega$ and step size $\alpha$. Then either for every $\alpha$ it is $y(\omega_k) = y(\omega)$ for all $k \in \mathbb{N}$, or there is some $\alpha_{\max} > 0$ such that for every $\alpha < \alpha_{\max}$ there is $n \in \mathbb{N}$ such that $y(\omega_k) = y(\omega)$ for all $k < n$ and $y(\omega_n) \neq y(\omega)$.*

*Proof.* Let us define $w\colon [0,\infty) \to W$ and $\gamma\colon [0,\infty) \to Y$ as

$$w(\alpha) = \omega - \alpha\Delta^{\mathrm{l}}\omega \quad \text{and} \quad \gamma(\alpha) = y\big(w(\alpha)\big) \quad \text{for } \omega \in W, \tag{45}$$

respectively. As $\gamma$ is a composition of an affine function $w$ and a piecewise constant solver, it is itself a piecewise constant function. Therefore, $\gamma$ induces a partitioning of its domain $[0,\infty)$ into disjoint sets on which $\gamma$ is constant. In fact, these sets are intervals, say $I_1, \ldots, I_m$, as intersections of the line segment $\{w(\alpha) : \alpha > 0\}$ and convex cones $W_y$. Consequently, $m \leq |Y|$.

Now, If $m = 1$, then $I_1 = [0,\infty)$ and $y(\omega_k)$ is constant $y(\omega)$ for all $k \in \mathbb{N}$ whatever $\alpha > 0$ is. In the rest of the proof, we assume that $m \geq 2$. Assume that the intervals $I_1, \ldots, I_m$ are labeled along increasing $\alpha$, i.e. if $\alpha_1 \in I_i$ and $\alpha_2 \in I_j$ then $\alpha_1 < \alpha_2$ if and only if $i < j$.

We define the upper bound on the step size to $\alpha_{\max} = |I_2|$. Assume that $\alpha < \alpha_{\max}$ and $(\omega_k)_{k=1}^\infty$ is given. Let $n = \min\{k \in \mathbb{N} : w(\alpha k) \in I_2\}$, i.e. the first index when the sequence $y(\omega_k)$ switches. Clearly $y(\omega_k) = \gamma(\alpha k) = y(\omega)$ for $k = 0, \ldots, n-1$ and $y(\omega_n) = \gamma(\alpha n) \neq y(\omega)$. □

**Proposition 3.** *Let $\alpha > 0$ and $(\omega_k)_{k=0}^\infty$ be as in* (35)*. Assume that $y(\omega_k) = y(\omega)$ for all $k < n$ and $y(\omega_n) \neq y(\omega)$. Also assume that from $\langle y(w_k), \omega_k \rangle = \langle y(\omega_{k-1}), \omega_k \rangle$ it follows that $y(w_k) = y(\omega_{k-1})$. Then*

$$f\big(y(\omega_n)\big) < f\big(y(\omega)\big), \tag{46}$$

*where $f$ is the linerarized loss at $y(\omega)$.*

*Proof.* As $y(\omega_k) = y(\omega)$ for all $k < n$, it is

$$\omega_k = \omega - n\alpha\Delta^\mathrm{I}\omega = \omega + n\alpha\frac{\mathrm{d}\ell}{\mathrm{d}y}. \tag{47}$$

Therefore, as $y(\omega_n)$ is the minimizer for the cost $\omega_k$, we have

$$0 \leq \langle y(\omega) - y(\omega_n), \omega_n \rangle = \langle y(\omega) - y(\omega_n), \omega \rangle + n\alpha\Big\langle y(\omega) - y(\omega_n), \frac{\mathrm{d}\ell}{\mathrm{d}y}\Big\rangle. \tag{48}$$

Now, since $y(\omega)$ is the minimizer for $\omega$, it is $\langle y(\omega) - y(\omega_n), \omega \rangle \leq 0$ and therefore

$$0 \leq \Big\langle y(\omega) - y(\omega_n), \frac{\mathrm{d}\ell}{\mathrm{d}y}\Big\rangle. \tag{49}$$

Consequently,

$$f\big(y(\omega_n)\big) = \ell\big(y(\omega)\big) + \Big\langle y(\omega_n) - y(\omega), \frac{\mathrm{d}\ell}{\mathrm{d}y}\Big\rangle \leq \ell\big(y(\omega)\big) = f\big(y(\omega)\big). \tag{50}$$

Equality is attained only if

$$\langle y(\omega_n) - y(\omega_{n-1}), \omega_n \rangle = 0. \tag{51}$$

This together with $y(\omega_n) \neq y(\omega) = y(\omega_{n-1})$ violates the assumption that the solver ties are broken in favour of the previously attained solution, therefore the inequality in (50) is strict. □

*Proof of Theorem 2.* Assume $Y^*\big(y(\omega)\big)$ is non-empty. It follows from Proposition 1 that there exists $m \in \mathbb{N}$ such that $y(\omega_m) \neq y(\omega)$. It follows from Proposition 2 that there is some $\alpha_{\max} > 0$ such that for every $\alpha < \alpha_{\max}$ there is $n \in \mathbb{N}$ such that $y(\omega_k) = y(\omega)$ for all $k < n$ and $y(\omega_n) \neq y(\omega)$. From Proposition 3 it also follows that $f\big(y(\omega_n)\big) < f\big(y(\omega)\big)$. Combining these results we have $y(\omega_n) \in Y^*\big(y(\omega)\big)$ by definition. This proves the first part of the theorem.

We prove the second part of the theorem by contradiction. Assume that $Y^*\big(y(\omega)\big)$ is empty and there exists some $\alpha$ such that $y(\omega_k) = y(\omega)$ for all $k < n \in \mathbb{N}$ and $y(\omega_n) \neq y(\omega)$. From Proposition 3 we know that $f\big(y(\omega_n)\big) < f\big(y(\omega)\big)$ and therefore $y(\omega_n) \in Y^*\big(y(\omega)\big)$. This is in contradiction to $Y^*\big(y(\omega)\big)$ being empty. □

## D  ADDITIONAL ILLUSTRATIONS

In addition to the illustrations for the 'simple' case in Fig. 2, we provide additional illustrations for the more challenging case in Fig. 9, in which the incoming gradient does not point towards an attainable solution. The illustrations compare the resulting update directions for Identity without projection, Identity with $P_{\mathrm{std}}$ projection, and Blackbox Backpropagation.

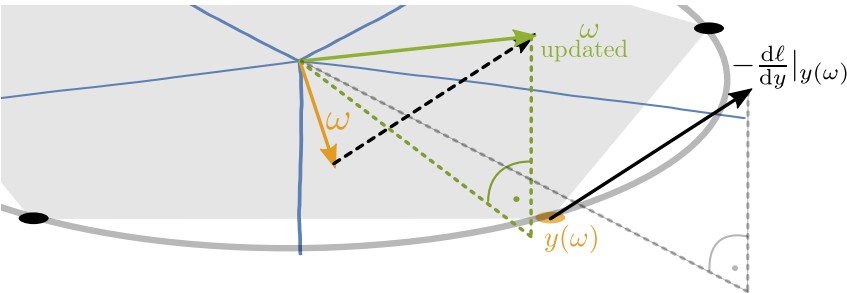

(a) Id update without projection: A large part of the resulting update does not affect the optimal solution. Note that the illustration shows a two-dimensional sphere embedded in a third dimension, with the incoming gradient pointing into the third dimension.

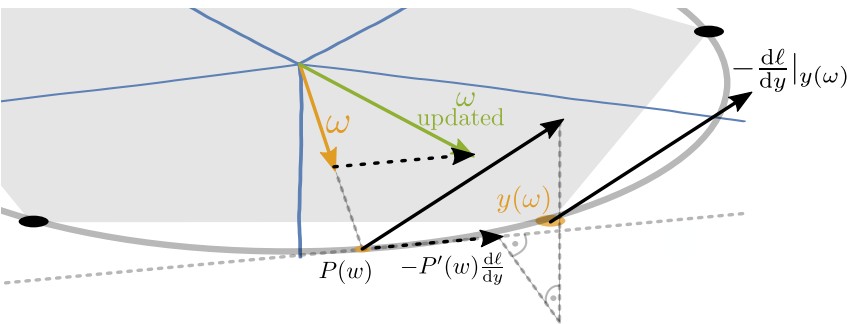

(b) Id update with projection $P_{\mathrm{std}}$: The invariant part of the update is removed by differentiating through the projection. It has the effect of first projecting onto the hyperplane, and then projecting onto the tangent of the sphere.

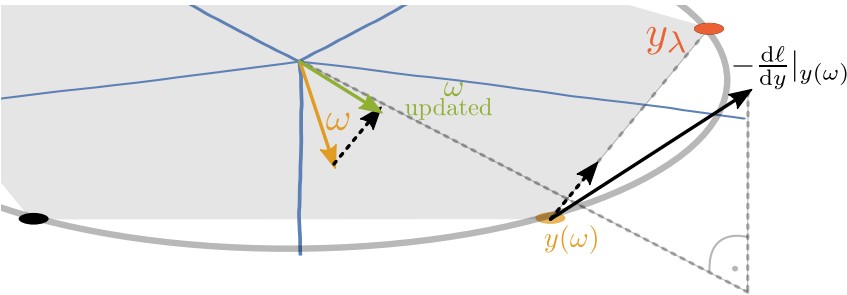

(c) BB update: The invariant part of the update is also removed, at the cost of an additional call to the solver. Shown is an update resulting from an appropriate choice of $\lambda$, which is in general not available a priori.

Figure 9: Intuitive illustration of the Identity (Id) gradient and Blackbox Backpropagation (BB) gradient when $-\mathrm{d}\ell/\mathrm{d}y$ does not point directly to a target. Illustrated is the case of differentiating an argmax problem. The cost and solution spaces are overlayed; the cost space partitions resulting in the same solution are drawn in blue. Note that the direct updates to the cost vector are only of illustrative nature, as the final updates are typically applied to the weights of the backbone.

