# OpenReview forum: "Backpropagation through Combinatorial Algorithms: Identity with Projection Works"
_ICLR.cc/2023/Conference — ICLR 2023 poster_

### Official Review · Reviewer_NUvG · 2022-10-21

**Confidence:** 3
**Correctness:** 4
**Technical Novelty And Significance:** 2
**Empirical Novelty And Significance:** 2
**Recommendation:** 3

**Clarity, Quality, Novelty And Reproducibility:**

**Contribution**

From what I can tell the main contribution is this: 1) if it is known that certain updates do not change a solution quality, then the component of the gradient update in this direction can be safely removed without hurting the loss by using $P \cdot (dl/dy)$, instead of $dl/dy$, and 2) actually it is a good thing to remove this redundant when using an adaptive optimizer like Adam since otherwise the learning rate will be adjusted downwards due to the spuriously big gradient size.

This is an interesting point, and is intuitive yet novel so far as I am aware. But the paper doesn't focus enough attention on this detail. There re a number of experiments showing the benefits of the projection, but no attempt is made to empirically validate the connection to adaptive training - it would be great to see plots showing the gradient update sizes with and without the projection, and plots showing the learning rate size and so on and so forth. Another interesting experiment would be to run training using non-adaptive SGD and show that there is no difference between the projection and no-projection methods. This would be good evidence to suppose the stated motivation for the projection method.

**Clarity**

- the paper is sufficiently clear so as to be quite quickly understood

**Reproducibility**

- the authors commit to releasing code when the paper is accepted. This is adequate in my view.

**Strength And Weaknesses:**

**Strengths**

- Paper is nicely written, an fairly clear
- A number of experimental setups are considered, I am happy to see a good variety of problems considered.
- The method is computationally cheap, and general purpose
- The argument that the identity is guaranteed not to find worse solutions is solid (e.g., Theorem 1), although of course the "there exists. gradient step such that the loss is better" doesn't say anything about how long before that useful step happens.

**Weaknesses**
- the identity method does appear to be very similar to the straight through estimator. Efforts are made in the paper to explain the difference but I currently fail to understand the differences.
- the projection idea is discussed in detail, and is motivated via the connection to adaptive methods. But no experiments are provided empirically validating this connection.

**Summary Of The Paper:**

This paper considers the problem of inserting combinatorial solvers into neural network architectures. Since combinatorial solvers have discrete outputs the gradient is zero or non-existent, meaning that an alternative "gradient" must be used on the backwards pass in order to provide directional information to update the model parameters.

This paper proposes a method that uses the identity function for the gradient, with an additional projection matrix applied in cases where these is a known invariance in the solution space. The projection is motivated by arguing that irrelevant gradient components interfere with adaptive methods such as Adam during training. This point is also empirically checked in several experiments. Other experiments show that the overall method is competitive and sometime better than representative methods in the related literature, e.g., the work of Vlastelica et al.

**Summary Of The Review:**




My main concern for rebuttal is this: can the authors please provide further clarification on the differences between their identity method and the Straight-Through estimator? I appreciate the efforts given in Appendix A to explain the differences, but unfortunately I still do not see what the difference is? For instance, Figure 5, which is used to explain the differences instantiates the proposed method for differentiation through a sampler. But the only non-differentiable part is the argmax, which uses the identity. Isn't this exactly what the straight through estimator would use too? I am very keen not to mis-accuse this work of being "the same" as the straight through estimator, but until I understand the difference I cannot advocate acceptance.

This paper has some interesting ideas, namely the projection method, but the emphasis of the paper is on the identity update, and secondarily on the projection. Consequently there is too little time spent discussing and studying the consequences of the projection idea. I really think this manuscript will make a nice paper eventually, but my suggestion to the authors would be to focus on the projection idea and the connection to adaptive methods. If you can show in more detail that this connection exists and is important to performance then this could be really helpful to practitioners.

Although I see a lot of promise in this work, in it's current state I cannot advocate for acceptance. I will pay very close attention to your rebuttal to make sure that I haven't made some terrible misunderstanding about the identity idea vs the straight through estimator.

---

> ### Author Response · Authors · 2022-11-15
> **Response to Reviewer NUvG - Re: Identity vs STE**
>
> We thank the reviewer for the constructive feedback and for raising questions that help us improve the paper.
>
> - Comparison to STE:
>     - We agree with the reviewer that the connection to the closely connected STE was not discussed in enough detail. In the updated version, we include an extensive discussion in supplementary A including the relevant references. We also modified the text in the main paper to reflect this by adding a paragraph in the related work (Section 2). In summary, identity generalizes STE to any argmax problem with a linear objective but gives different updates compared to previous STE generalizations. A compressed version of the full discussion is given here:
>     - History of the STE:
>         - Originally, the STE refers to differentiating the hard thresholding function $I_{x\ge 0}$, by replacing it with the identity function on the backward pass. [1, 2]
>         - The thresholding function can also be used to model sampling from a Bernoulli distribution (see manuscript for details) [3]. Using the Identity as a proxy for differentiation of the thresholding function can be interpreted as using the expected value of the random Bernoulli variable as a proxy for differentiating through a discrete sample.
>         - Interestingly, this connection translates to sampling from other discrete distributions, which gives rise to a generalization of the STE. As an example, consider the case of sampling from a discrete exponential family distribution over one-hot vectors. Here, the expected value (which is the softmax) was used as a proxy for differentiating through the discrete samples. This instance of the STE was called Gumbel STE (for the reasoning behind this, see discussion in manuscript) [4].
>         - In general, however, computing the expected value can be intractable, as exponentially many structures might need to be considered. Therefore, this generalization of the STE is rather limited.
>     - Connection of Identity to STE:
>         - The original purpose of the STE, i.e. differentiating the hard thresholding function, can also be recovered with our method, by formulating the hard thresholding function as a discrete optimization problem (see manuscript for details). In this sense, the Identity method also generalizes the STE to linear optimization problems over more general solution sets (rather than the simple $\{0,1\}$-set).
>         - However, this form of generalization differs from the previously used generalizations of the STE, which is especially visible in cases that allow the application of both methods. As an example, consider again the discrete sampling from the set of one-vectors described above. The sampling process can be also formulated as a stochastic argmax problem, and is therefore amenable to the Identity method. The vanilla identity method uses the identity as a proxy for differentiation, in contrast to the previously described STE generalization, which differentiates through the softmax, as described above. The benefit of the simplicity of the Identity method is that it can be applied to any (stochastic) argmax problem, without potentially intractable computations of expectations.
>         - Note: The STE has also been used to refer to cases in which any computational block is replaced with an identity proxy on the backward pass [3, 5]. In this much looser context, our method falls in the category of STEs, however, we believe that additionally overloading the term would not benefit the community.
>
> [1]: Frank Rosenblatt. The perceptron: A probabilistic model for information storage and organization in the brain. Psychological Review, 1958.
>
> [2]: Geoffrey Hinton. Neural networks for machine learning, coursera. Coursera, video lectures, 2012.
>
> [3]: Yoshua Bengio, Nicholas Léonard, and Aaron C. Courville. Estimating or propagating gradients
>
> through stochastic neurons for conditional computation. CoRR, 2013.
>
> [4]: Eric Jang, Shixiang Gu, and Ben Poole. Categorical reparameterization with gumbel-softmax. ICLR 2017
>
> [5]: Penghang Yin, Jiancheng Lyu, Shuai Zhang, Stanley J. Osher, Yingyong Qi, and Jack Xin. Understanding straight-through estimator in training activation quantized neural nets. ICLR 2019.

---

> ### Author Response · Authors · 2022-11-15
> **Response to Reviewer NUvG - Re: Connection to adaptive optimizers**
>
> We thank the reviewer for the constructive feedback and for raising questions that help us improve the paper.
>
> - Connection to adaptive optimizers:
>     - We would like to clarify that the connection to adaptive optimizers was not the main motivation behind introducing the projections. Instead, the motivation was the observation that the gradient computed by the vanilla Identity method often points in the ‘spurious direction’ (i.e. in the case discussed in Section 3, illustrated in Fig. 9), when compared to, e.g., Blackbox Backpropagation. The argument that this can potentially cause problems with adaptive optimizers was more of a hypothesis of the consequence of spurious update directions.
>     - To appropriately test this hypothesis, we now evaluated the Identity method with the SGD optimizer, and we did not observe a major change in performance compared to adaptive optimizers, see Fig. 6 (supplementary B.1). Accordingly, we removed it from the main text paper and only keep the comparison in the supplementary.
>     - However, problems with adaptive optimizers are just one possible consequence of spurious gradient directions. To see why this is the case, it is very important to note that the final updates are actually applied to the parameters of the neural network that predicts the cost vector, and not directly in the cost space as illustrated in Fig. 2 and Fig. 6 (supplementary D). For example, if in Fig. 6a updates were performed directly in cost space, one could argue that the final update to the cost vector still has a ‘relevant’ component, and as the ‘irrelevant’ component does not matter for the solver, repeated updates will eventually make the solver switch to the potentially better solution. However, as we perform updates in the much more complicated space of the neural network parameters, the noise from the irrelevant part of stochastic samples can easily overshadow any relevant information. We still consider the illustrations to be helpful for the intuition, however, we will add a disclaimer about the misleading intuition described above.
>     - To experimentally verify the statement, that relevant information can be lost in the stochasticity of the irrelevant part, we compared the gradients before and after the projection in the DVAE experiment. As reported in Fig. 7 (supplementary B.1) we observe that a very large part of the gradient is removed by the projection and the gradient direction is significantly altered (cosine similarity < 0.05). Given that with projection the loss is optimized well, we can conclude that relevant information is indeed retained whereas spurious directions are removed.
>
> [1]: Frank Rosenblatt. The perceptron: A probabilistic model for information storage and organization in the brain. Psychological Review, 1958.
>
> [2]: Geoffrey Hinton. Neural networks for machine learning, coursera. Coursera, video lectures, 2012.
>
> [3]: Yoshua Bengio, Nicholas Léonard, and Aaron C. Courville. Estimating or propagating gradients
>
> through stochastic neurons for conditional computation. CoRR, 2013.
>
> [4]: Eric Jang, Shixiang Gu, and Ben Poole. Categorical reparameterization with gumbel-softmax. ICLR 2017
>
> [5]: Penghang Yin, Jiancheng Lyu, Shuai Zhang, Stanley J. Osher, Yingyong Qi, and Jack Xin. Understanding straight-through estimator in training activation quantized neural nets. ICLR 2019.

---

> ### Author Response · Authors · 2022-12-01
> **Follow-up on Rebuttal**
>
> We again thank the reviewer for the time and expertise invested in the review. We would like to kindly ask the reviewer to let us know whether our answers and changes to the manuscript have clarified the raised concerns, and whether this changes the reviewers evaluation of the paper. Best regards,
> the authors

---

### Official Review · Reviewer_MfHU · 2022-10-24

**Confidence:** 4
**Correctness:** 4
**Technical Novelty And Significance:** 4
**Empirical Novelty And Significance:** Not applicable
**Recommendation:** 6

**Clarity, Quality, Novelty And Reproducibility:**

The paper is clearly written, and proposes a simple new way of providing an update to the cost vector of a combinatorial problem that aims at achieving lower loss. Ample information for reproducibility is provided in the appendix.

**Strength And Weaknesses:**

Strengths:
- simplicity and computational efficiency of the "negative identity" approach compared to e.g. Vlastelica's approach, which requires two calls to the solver per gradient iteration
- consideration for solver invariants is a plus
- empirical evaluation on a wide range of problems is a plus

Weaknesses:
- intuition and theory in 3.2 and 3.3 shows that the update leads to a lower-loss solution. It would be helpful to have some estimate/indication of how does the update compare with the best possible update (of some limited magnitude)
- empirical evaluation is rather limited in terms of competing methods. It would be interesting to see how the proposed method compares with e.g. cvxpylayers and similar approaches on combinatorial problems equivalent to convex optimization problems, where these methods can be applied
- it would be helpful to have some complexity/run-time information provided for all the methods/runs in Section 4, to allow for judging the tradeoff between accuracy and speed.
- the relative comparisons show somewhat mixed results, e.g. no improvement over Softsub and small over I-MLE in Table 1, similar results to BB in Table 2, lower performance than BB in Table 3, similar performance but higher robustness than BB in Fig. 4.
- minor: graphical illustration for the more challenging case (Section 3.3) would increase the accessibility of the paper.

**Summary Of The Paper:**

The paper proposes a surrogate gradient for differentiating through combinatorial solvers with linear objective function, for which gradient is not informative (either null or does not exist).

**Summary Of The Review:**

The paper presents a novel way to perform updates over combinatorial solvers. The approach is simpler compared to prior approaches, yet in some cases empirically competitive. While well-argued and well-written, the paper could benefit from a more in-depth theoretical analysis, and from more information in the experimental section.

---

> ### Author Response · Authors · 2022-11-15
> **Response to Reviewer MfHU - Re: In-depth Theoretical Analysis**
>
> We thank the reviewer for their helpful feedback on the manuscript.
>
> - Concern 1: The paper requires more in-depth theoretical analysis (comparison of update with best possible update, graphical illustration of the more challenging case).
>     - In general, when differentiating through discrete solver, is not entirely clear what the best possible update is. In theory, the ultimately best update would be to compute the globally optimal solution $y_\text{best}=\min_{y\in Y} \ell(y)$ which has the minimum loss of all possible solutions in $Y$. In general, although possible to solve approximately, this is a much more expensive computation than the linear program (LP) solved on the forward pass.
>     - To alleviate this, one can stay true to the spirit of first-order methods by instead minimizing the linearized loss, i.e. $y_\text{best}=\min_{y\in Y} \langle y, \frac{d\ell}{dy}\rangle$. This amounts to simply solving another LP on the backward pass, however, the linearization of the loss is only truthful to the true loss in a region around the current optimal solution at which we linearize it.
>     - This is the reason why Blackbox Backpropagation (BB) computes the target by solving $y_\lambda=\min_{y\in Y} \langle y, \omega+\lambda\frac{d\ell}{dy}\rangle$, which involves both the current cost vector $\omega$ (to preserve locality), and the loss linearization. In this sense, the target $y_\lambda$ computed by BB is somewhat optimal considering the limited computation budget, as long as the correct value for $\lambda$ is selected (i.e. the smallest value such that $y_\lambda$ is different from the current solution).
>     - Finally, after deciding on a target $y_\lambda$, BB uses the update $\Delta^\text{BB}\omega\propto y_\lambda-y(\omega)$. This is arguably the best possible update to in order to get from the current $y(\omega)$ to $y_\lambda$, as the update is orthogonal to the hyperplane $H=\{\omega\in\mathbb{R}^n:\langle y_\lambda-y(\omega),\omega\rangle=0\}$ that separates the two solutions in cost space.
>     - From the perspective described above, the BB update can reasonably be considered the best possible update. The manuscript contains various comparisons to BB, and we further extended this with additional graphical illustrations in supplementary D.

---

> ### Author Response · Authors · 2022-11-15
> **Response to Reviewer MfHU - Re: More information in the Experimental Section**
>
> We thank the reviewer for their helpful feedback on the manuscript.
>
> - Concern 2: Paper requires more information in the experimental section (comparison with existing methods, runtimes), and the proposed method leads to mixed results.
>     - Comparison to existing methods:
>         - The problems considered in our experiments could also be formulated in cvxpy-layers by adding an entropic regularizer to allow differentiation. However, this goes against the paradigm of blackbox differentiation which enables the user to effortlessly choose any blackbox solver optimized for the current task, without any modifications, such as relaxations or costly non-linear regularizations.
>         - In all our conducted experiments we compare to representatives of the current state of the art methods. At this point, these do not include approaches involving cvxpy-layers for the experiments we are considering. We agree that examining whether differentiating a relaxed optimization problem with cvxpy is competitive on these benchmarks would be an interesting investigation, however, we believe that evaluating additional methods besides the state-of-the-art is not essential to demonstrate the efficiency of our proposed simplistic method.
>     - Runtimes:
>         - We added a table with runtime comparisons to Table 8 (supplementary B.6). We observe, as expected, that the benefit from saving a second invocation of the solver is most prominent in the case of more difficult optimization problems, most prominently in the NP-hard TSP problem. In theory, the complexity of the Identity method is half of that of Blackbox Backpropagation (BB), as it removes the need for an additional solver invocation on the backward pass.
>     - Mixed results:
>         - We would like to clarify that the aim of the paper is not to improve upon the performance of state-of-the-art methods but to show the circumstances under which a more lightweight computation for an informative gradient can produce comparable results.
>         - In the experiments, we show that in the simple (yet common) case of a loss defined directly on the solver solution, we indeed match the performance of competing methods. In the more challenging general case, we observe the theoretically anticipated difficulties from naively applying the vanilla identity method, and show how the proposed projections can reduce the drawbacks. However, throughout the experimental evaluation, we also highlight the limitations that the identity method has due to its simplicity, which hopefully allows readers to choose the correct method for the problem at hand.
>         - Note: While running the ablations for the L2X experiment requested by Reviewer bT3d, we discovered an inconsistency in the code for L2X. We now report the new results after fixing the bug, which are better than the previously reported numbers and outperform Softsub for larger $k$. However, superior performance is not the main benefit of our method.

---

> ### Comment · Reviewer_MfHU · 2022-11-18
> **post rebuttal**
>
> I'm satisfied with authors' response, and I'm raising my score from 5 to 6.

---

### Official Review · Reviewer_bT3d · 2022-10-25

**Confidence:** 3
**Correctness:** 3
**Technical Novelty And Significance:** 3
**Empirical Novelty And Significance:** 3
**Recommendation:** 6

**Clarity, Quality, Novelty And Reproducibility:**

The paper said: a curated Github repository for reproducing all other results will be published upon acceptance.

**Strength And Weaknesses:**

## Pros
- The motivation of this work is clear and the proposed method seems simple and efficitive. It is suprising that the identity method, which is similar to Straight-through estimator for samples drawn, can be applied to optimization operation.
- Extensive experimental results on discrete samplers, deep graph matching, image retrrival, etc. show the effectiveness of the proposed method.

## Cons
- The paper is not easy to read and some place is hard to decipher. Some derivations are not very clear due to omissions or inappropriate notation usage：
1. I think  that Equation (2) should be something like $\frac{d}{d\omega}\langle\omega, y^* - y(\omega^0)\rangle$ where $y(\omega^0)$ is a fixed vector rather than a function of $\omega$.
2. In Equation (6), the linearization $f(y)$ can be viewed as the Taylor expansion of $l(y)$ around $y(\omega)$, which means $f(y)$ can only approximate $l(y)$ in the small region around $y(\omega)$. However, for the more wide region $y \in Y$, a lower linearized loss doesn't mean a lower true loss. For instance, assume $y \in \mathbb R^2$ and the feasible region is the polygon with extreme points $\{(0, -1), (0,10),(-10,0), (10,0)\}$, and the nonlinear loss $l(y) = ||y||^2$. If $y(\omega) = (0, -1)$, by the defination of Equation (6) $Y^*(y(\omega)) = \{(0,10),(-10,0), (10,0)\}$. All points in this set have a lower linearized loss but higher true loss than $y(\omega)$.
3. In the Linear Transforms part of Section 3.4, $\Delta^I\omega$ is decomposed into $\Delta^I\omega = \Delta^I\omega_1 + \Delta^I\omega_2$, where $\Delta^I\omega_1 = P\Delta^I\omega, \Delta^I\omega_2 = \Delta^I\omega - \Delta^I\omega_1$. It is clear if $P$ is a projection matrix (e.g. the $P_{mean}$ in Equation (13)), then $\Delta^I\omega_2 \in \text{ker } P$. But for general $P$, which is not a projection matrix, whether $\Delta^I\omega_2 \in \text{ker } P$ still holds? This may affect the derivation of Nonlinear Transforms.
- Some additional ablation studies about invariant mappings should be conducted: In Table 1 and Table 3, $P_{norm}$ cooperate with $P_{mean}$ to improve the performance. In Table 2 and Table 4, the performance of Identity with $P_{norm}$ alone is worse than that without projection. Therefore, the ablation study is needed to find out whether $P_{norm}$ is useful. For example, in Table 1, besides $Id. (P_{std})$ and $Id. (\text{no } P)$, $Id. (P_{mean})$ and  $Id. (P_{norm})$ should also be evaluated to find out whether these two mappings alone help to improve model performance.

**Summary Of The Paper:**

This paper proposes a straightforward hyper-parameter-free approach to embed discrete solvers as differentiable layers into neural networks. In detail, during the backward pass, the Jacobian $\frac{\partial y(\omega)}{\partial \omega}$ is simply treated as a negative identity matrix and a theoretical justification is provided. As the identity method alone can result in unstable learning behavior, two transformations (i.e. mean and norm) are applied to the cost vector $\omega$ to exploit invariants. Noise is added to the cost vector to improve the robustness. Experiments on discrete samplers, deep graph matching, and image retrieval show the proposed method is competitive against previous more complex methods.

**Summary Of The Review:**

I would to reconsider my rating based upon authors' feedback and clarification. Currently it is hard for me to fully understand the paper.

---

> ### Author Response · Authors · 2022-11-15
> **Response to Reviewer bT3d**
>
> We thank the reviewer for their detailed read and useful feedback on the manuscript.
>
> - Main concern of the reviewer: Parts of the paper are difficult to decipher/unclear.
>     1. In Eq. (2), the term $y(\omega)$ is indeed a function of $\omega$. The mapping $\omega\mapsto y(\omega)$ is a piecewise constant function and hence the derivative of the term $\langle\omega,y(\omega)\rangle$ equals $y(\omega)$ except at the points where $y$ has jumps (and derivative does not exist here).
>     2. We agree with your observation. This a general drawback of all first-order (gradient-based) methods, that a gradient provides only *local* information and is not able to capture the *global* behavior. (Note that the gradients of the true loss $\ell$ and its linearization—first-order Taylor expansion— $f$ at a given point, are the same). This is also the reason why backpropagation through solvers is difficult: The incoming gradient $-\nabla_y\ell$ tells us in which direction the loss decreases the most and this information is meaningful only in a small neighborhood of a given point. However, from the solver’s perspective, this local information is meaningless (solution $y$ does not change with small changes to the cost $\omega$) and the global information is (as you pointed out) detached from what we *actually* want to optimize. The goal is to find a reasonable trade-off (cf. also the BB paper [1], p. 4, where this is also discussed and showed how the parameter $\lambda$ controls this trade-off.) Therefore this is an established and fundamental challenge in this area and *not a drawback* of Identity or any other first-order method.
>     3. The composition of a transform $P$ and the solver (argmin) leads to the equation (10) simply by the chain rule (with argmin treated as identity). The linear case serves rather as an illustrative example and not as a derivation. To see that it is in agreement with the general case, we use that $P$ behaves *locally* as its linear approximation. Then, $P'$ is the linear object (projection matrix) that was discussed first (therefore $\Delta^I\omega_2\in\ker P'$, and not $P$).  We clarified this in the revised version (see Section 3.4), and additionally illustrated the use of $P'$ instead if $P$ in in Fig. 9b (supplementary D).
> - Additional concern: Ablation studies about invariant mappings.
>     - We agree that separating the contributions of the individual projections is important, and therefore added the requested ablations to assess the effect of $P_\text{norm}$ and $P_\text{mean}$ individually in the experiments where we used $P_\text{std}$, i.e. top-$k$ in the L2X experiment (Table 1) and ranking in CUB experiment (Table 3). The new results, reported in supplementary B (Fig. 5 and Table 5), show that both projections increase the performance individually (with a much larger effect of $P_\text{norm}$ than $P_\text{mean}$), but combined they perform the best. This is intuitively understandable in terms of viewing projections as solver relaxations (as now described in more detail in Section C.2): No projection, mean projection, normalization, and standardization correspond to relaxations of $Y$ to $\mathbb R^n$, a hyperplane $H$, a hypersphere $S$, and the intersection $S\cap H$, respectively, which increasingly better approximate the true $Y$.
>
> [2]: Marin Vlastelica, Anselm Paulus, Vít Musil, Georg Martius, and Michal Rolínek. Differentiation of blackbox combinatorial solvers. ICLR 2020

---

> > ### Comment · Reviewer_bT3d · 2022-11-18
> > **score increase**
> >
> > The authors' response addressed most of my concerns and the additional ablation study on contributions of individual projections makes the work more complete. I'm happy to raise my score to 6 now.

---

### Official Review · Reviewer_cnD6 · 2022-11-03

**Confidence:** 4
**Correctness:** 4
**Technical Novelty And Significance:** 3
**Empirical Novelty And Significance:** 3
**Recommendation:** 8

**Clarity, Quality, Novelty And Reproducibility:**

The writing of the paper is good. I feel very pleased and smooth reading this paper.
Novelty of this paper is also above the average of ICLR.
The authors also provided the code for reproducing part of the experiments.

**Strength And Weaknesses:**

In general, I like reading this paper since it was well-written and very easy to follow. All the necessary details were presented and the idea was insightful. To me, the strength is as follows:
1. The idea of utilizing identity as a gradient estimator for linear combinatorial solvers is very interesting and neat. This is without any extra calling of the solvers and can be easily extended to a large variety of downstream tasks.
2. Although the technical part of the proof of Theorem 1 is not so complicated, I think Theorem 1 can support the claims about the applicability suggested by the authors.
3. The authors provided an initial but insightful discussion on utilizing projection as solver relaxation.

Despite the aforementioned strength, this paper can be further improved:
1. The discussion of projection as solver relaxation seems not so thorough and solid. More detailed and theoretically guaranteed discussion can greatly strenghened the contribution of this paper.
2. There have been several previous works approximating gradient of linear combinatorial solvers. I suggest the authors to give some discussion on the non-linear cases (say, quadratic).
3. Aside from using bar graph in Table 2 and 3, it could be more readable if numbers could be provided.


**Summary Of The Paper:**

This paper proposed a principled method to approximate the gradient of a combinatorial solver, which could be readily integrated into an end-to-end learning pipeline. To achieve this, the authors studied the behavior of linear combinatorial solvers and proposed "Identity" as the surrogate gradient over the coefficients, with further added perturbation to avoid the cost collapse. Experiments were conducted on various combinatorial tasks, showing its superior performance over existing gradient estimators. The idea of utilizing identity as a gradient is interesting and novel, with some theoretical guarantee.

**Summary Of The Review:**

In general, I vote an acceptance for this paper, taking into account its readability, novelty, theoretical soundness and empirical performance.

---

> ### Author Response · Authors · 2022-11-15
> **Response to Reviewer cnD6**
>
> We thank the reviewer for their constructive feedback on the manuscript.
>
> - Discussion of projection as solver relaxation:
>     - We improved the presentation by adding a more detailed discussion of the connections between projections in cost space and relaxations in solution space to supplementary C.2, where we connect each of the used projections to a corresponding relaxation (we now also include the vanilla identity method without relaxations, and the projection onto a hyperplane). We also updated the main text in Section 3.5 to reflect this.
> - Discussion of the non-linear case:
>     - In contrast to intuition, the non-linear convex case is in fact a simpler problem to differentiate than the linear case, which is treated in this paper, due to the inherent discreteness of the solution set. Considering the case of an entropic term in the objective (which blows up at the boundary of the polytope), the optimal solution is always located in the interior of the feasible set.
>     Intuitively, this means that there exists a continuously differentiable relationship between the objective parameters and the optimal solution. The corresponding non-zero Jacobian can be calculated by solving the KKT system on the backward pass, see e.g. OptNet [1] for a detailed discussion.
>     In contrast, for an LP the optimal solution is always located at one of the extremal points of the feasible region, therefore there is a discontinuous relationship between the cost vector parameters and the optimal solution, which requires the use of a surrogate Jacobian to replace the true zero Jacobian.
>     Quadratic programs are situated somewhere between the extreme cases discussed above, as both scenarios (optimal solution at extremal point or not) are possible, depending on the strength of the regularizer. Existing methods such as OptNet typically just compute the true zero Jacobian in the case of an extremal solution, and rely on the signal from other less problematic samples in the stochastic optimization of the full model.
>     Note: We added a short discussion of the described relationship to Section 3.1 and supplementary C.1 in the manuscript.
> - Providing numbers:
>     - We added numbers to supplement the visual presentation in supplementary B.2 (Table 4) and B.3 (Table 5).
>
> [1]: Brandon Amos and J. Zico Kolter. Optnet: Differentiable optimization as a layer in neural networks. ICML 2017

---

> > ### Comment · Reviewer_cnD6 · 2022-11-18
> > **response to the authors**
> >
> > Thank you for providing timely feedback. The revised version seems more friendly to me.
> >
> > I believe this is a good piece of work bringing about insights into related problems. I thus vote an acceptence.

---

### Author Response · Authors · 2022-11-15
**Response to all reviewers**

We thank all the reviewers for their constructive feedback on the paper, which helped us to majorly improve parts of the manuscript. The revised version of the paper incorporates many of the provided suggestions (highlighted in blue), including

- a detailed comparison to the Straight-through estimator in the related work and supplementary A,
- additional ablations of splitting $P_\text{std}$ into $P_\text{mean}$ and $P_\text{norm}$ for the DVAE and CUB experiment, as reported and discussed in supplementary B.1 (Fig. 5) and supplementary B.3 (Tab. 5),
- a runtime comparison in supplementary B.6 (Tab. 8),
- a discussion of the differences between the linear case and the non-linear case in supplementary C.1,
- a more detailed discussion of the connection between projections and relaxations in supplementary C.2,
- additional illustrations of the update directions in supplementary D (Fig. 9),
- additional minor improvements to the presentation in the main text.

We hope that our revision addresses all the concerns raised by the reviewers, and we appreciate any additional feedback to further improve the paper.

---

### Comment · Area_Chair_pDvE · 2022-11-18
**Please respond to author rebuttals**

Dear Reviewers,

The authors have submitted their rebuttals. Please have a look and respond to their efforts. This will be a respect to their hard work. Many thanks!

Area Chair

---

### Decision · Program_Chairs · 2023-01-20

**Decision:**

Accept: poster

**Justification For Why Not Higher Score:**

The paper is surely at most a poster as it is indeed a borderline paper as all reviewers agreed.

**Justification For Why Not Lower Score:**

Anyway, three reviewers gave positive scores and only one was negative towards it. Moreover, the negative reviewer did not object accepting the paper.

**Metareview: Summary, Strengths And Weaknesses:**

The paper originally got one 8 (accept, good paper), two 5s (marginally below threshold) and one 3 (reject). The major challenges include insufficient theoretical guarantee on the proposed trick, missing some related works, being possibly difficult to read, missing additional ablation studies, etc. The author rebuttals were more or less successful and two reviewers raised from 5 to 6. However, during the virtual meeting, the reviewers did not fully agree on the difference between Idendity and Strate-Through Estimator but all agreed that the applicable scope of the Idendity+Projection is a bit limited. By the overall scores and discussion with the reviewers, the AC recommended acceptance, although not a strong one.

**Note From Pc:**

if the above contains the word "oral" or "spotlight" please see: "oral" presentation means -> notable-top-5% and "spotlight" means -> notable-top-25%. As stated in our emails, we are disassociating presentation type from AC recommendations

**Summary Of Ac-Reviewer Meeting:**

The key problem is whether the Identity is sufficiently different from the Straight-Through Estimator. The reviewers did not have an agreement on this problem, but they all agree that the proposed Identity+Projection method has limited applicable scope and the paper is not strong anyway. Finally, the reviewer who gave 3 did not object accepting the paper.